# The Provable Benefits of Unsupervised Data Sharing for Offline Reinforcement Learning

**Hao Hu**[1]* **Yiqin Yang**[2]* **Qianchuan Zhao**[2]**, Chongjie Zhang**[1]
[1]Institute for Interdisciplinary Information Sciences, Tsinghua University
[2]Department of Automation, Tsinghua University
`{huh22,yangyiqi19}@mails.tsinghua.edu.cn`
`{chongjie,zhaoqc}@tsinghua.edu.cn`

## Abstract

Self-supervised methods have become crucial for advancing deep learning by leveraging data itself to reduce the need for expensive annotations. However, the question of how to conduct self-supervised offline reinforcement learning (RL) in a principled way remains unclear. In this paper, we address this issue by investigating the theoretical benefits of utilizing reward-free data in linear Markov Decision Processes (MDPs) within a semi-supervised setting. Further, we propose a novel, **P**rovable **D**ata **S**haring algorithm (PDS) to utilize such reward-free data for offline RL. PDS uses additional penalties on the reward function learned from labeled data to prevent overestimation, ensuring a conservative algorithm. Our results on various offline RL tasks demonstrate that PDS significantly improves the performance of offline RL algorithms with reward-free data. Overall, our work provides a promising approach to leveraging the benefits of unlabeled data in offline RL while maintaining theoretical guarantees. We believe our findings will contribute to developing more robust self-supervised RL methods.

## 1 Introduction

Offline reinforcement learning (RL) is a promising framework for learning sequential policies with pre-collected datasets. It is highly preferred in many real-world problems where active data collection and exploration is expensive, unsafe, or infeasible (Swaminathan & Joachims, 2015; Shalev-Shwartz et al., 2016; Singh et al., 2020). However, labeling large datasets with rewards can be costly and require significant human effort (Singh et al., 2019; Wirth et al., 2017). In contrast, unlabeled data can be cheap and abundant, making self-supervised learning with unlabeled data an attractive alternative. While self-supervised methods have achieved great success in computer vision and natural language processing tasks (Brown et al., 2020; Devlin et al., 2018; Chen et al., 2020), their potential benefits in offline RL are less explored.

Several prior works (Ho & Ermon, 2016; Reddy et al., 2019; Kostrikov et al., 2019) have explored demonstration-based approaches to eliminate the need for reward annotations, but these approaches require samples to be near-optimal. Another line of work focuses on data sharing from different datasets (Kalashnikov et al., 2021; Yu et al., 2021a). Still, it assumes that the dataset can be relabeled with oracle reward functions for the target task. These settings can be unrealistic in real-world problems where expert trajectories and reward labeling are expensive.

Incorporating reward-free datasets into offline RL is important but challenging due to the sequential and dynamic nature of RL problems. Prior work (Yu et al., 2022) has shown that learning to predict rewards can be difficult, and simply setting the reward to zero can achieve good results. However, it's unclear how reward-prediction methods affect performance and whether reward-free data can provably benefit offline RL. This naturally leads to the following question:

*How can we leverage reward-free data to improve the performance of offline RL algorithms in a principled way?*

---

*Equal contribution.

To answer this question, we conduct a theoretical analysis of the benefits of utilizing unlabeled data in linear MDPs. Our analysis reveals that although unlabeled data can provide information about the dynamics of the MDP, it cannot reduce the uncertainty over reward functions. Based on this insight, we propose a model-free method named Provable Data Sharing (PDS), which adds uncertainty penalties to the learned reward functions to maintain a conservative algorithm. By doing so, PDS can effectively leverage the benefits of unlabeled data for offline RL while ensuring theoretical guarantees.

We demonstrate that PDS can cooperate with model-free offline methods while being simple and efficient. We conduct extensive experiments on various environments, including single-task domains like MuJoCo (Todorov et al., 2012) and Kitchen (Gupta et al., 2019), as well as multi-task domains like AntMaze and Meta-World (Yu et al., 2020a). The results show that PDS improves significantly over previous methods like UDS (Yu et al., 2022) and naive reward prediction methods.

Our main contribution is the Provable Data Sharing (PDS) algorithm, a novel method for utilizing unsupervised data in offline RL that provides theoretical guarantees. PDS adds uncertainty penalties to the learned reward functions and can be easily integrated with existing offline RL algorithms. Our experimental results demonstrate that PDS can achieve superior performance on various locomotion, navigation, and manipulation tasks. Overall, our work provides a promising approach to leveraging the benefits of unlabeled data in offline RL while maintaining theoretical guarantees and contributing to the development of more robust self-supervised RL methods.

## 2 RELATED WORK

**Offline Reinforcement Learning** Current offline RL methods (Levine et al., 2020) can be roughly divided into policy constraint-based, uncertainty estimation-based, and model-based approaches. Policy constraint methods aim to keep the policy close to the behavior under a probabilistic distance (Siegel et al., 2020; Yang et al., 2021; Kumar et al., 2020; Fujimoto & Gu, 2021; Yang et al., 2022; Fujimoto et al., 2019; Hu et al., 2022; Kostrikov et al., 2021; Ma et al., 2021; Wang et al., 2021). Uncertainty estimation-based methods attempt to consider the Q-value prediction's confidence using dropout or ensemble techniques (An et al., 2021; Wu et al., 2021). Last, model-based methods incorporates the uncertainty in the model space for conservative offline learning (Yu et al., 2020b; 2021b; Kidambi et al., 2020).

**Offline Data Sharing** Prior works have demonstrated that data sharing across tasks can be beneficial by designing sophisticated data-sharing protocols (Yu et al., 2021a). For example, previous studies have explored developing data sharing strategies by human effort (Kalashnikov et al., 2021), inverse RL (Reddy et al., 2019; Li et al., 2020), and estimated Q-values (Yu et al., 2021a). However, these data sharing works must assume the dataset can be relabeled with oracle rewards for the target task, which is a strong assumption since the high cost of reward labeling. Therefore, effectively incorporating the unlabeled data into the offline RL algorithms is essential. To solve this issue, some recent work (Yu et al., 2022) proposes simply applying zero rewards to unlabeled data. In this work, we propose a principle way to leverage unlabeled data without the strong assumption of reward relabeling.

**Reward Prediction** It is widely observed that reward shaping and intrinsic rewards can accelerate learning in online RL (Mataric, 1994; Ng et al., 1999; Wu & Tian, 2017; Song et al., 2019; Guo et al., 2016; Abel et al., 2021). There are also extensive works that studies automatically designing reward functions using inverse RL (Ng et al., 2000; Fu et al., 2017). However, there is less attention on the offline setting where online interaction is not allowed and the trajectories may not be optimal.

## 3 PRELIMINARIES

### 3.1 LINEAR MDPS AND PERFORMANCE METRIC

We consider infinite-horizon discounted Markov Decision Processes (MDPs), defined by the tuple $(\mathcal{S}, \mathcal{A}, \mathcal{P}, r, \gamma)$, with state space $\mathcal{S}$, action space $\mathcal{A}$, discount factor $\gamma \in [0, 1)$, transition function $\mathcal{P} : \mathcal{S} \times \mathcal{A} \rightarrow \Delta(\mathcal{S})$, and reward function $r : \mathcal{S} \times \mathcal{A} \rightarrow [0, r_{\max}]$. To make things more concrete,

we consider the *linear MDP* (Yang & Wang, 2019; Jin et al., 2020) as follows, where the transition kernel and expected reward function are linear with respect to a feature map.

**Definition 3.1** (Linear MDP). We say an episodic MDP $(\mathcal{S}, \mathcal{A}, \mathcal{P}, r, \gamma)$ is a linear MDP with known feature map $\phi : \mathcal{S} \times \mathcal{A} \to \mathbb{R}^d$ if there exist unknown measures $\mu = (\mu_1, \ldots, \mu_d)$ over $\mathcal{S}$ and an unknown vector $\theta \in \mathbb{R}^d$ such that

$$\mathcal{P}(s' \,|\, s, a) = \langle \phi(s, a), \mu(s') \rangle, \quad r(s, a) = \langle \phi(s, a), \theta \rangle \tag{1}$$

for all $(s, a, s') \in \mathcal{S} \times \mathcal{A} \times \mathcal{S}$. And we assume $\|\phi(s, a)\|_2 \leq 1$ for all $(s, a, s') \in \mathcal{S} \times \mathcal{A} \times \mathcal{S}$ and $\max\{\|\mu(\mathcal{S})\|_2, \|\theta\|_2\} \leq \sqrt{d}$, where $\|\mu(\mathcal{S})\| \equiv \int_{\mathcal{S}} \|\mu(s)\| ds$.

A policy $\pi : \mathcal{S} \to \Delta(\mathcal{A})$ specifies a decision-making strategy in which the agent chooses actions adaptively based on the current state, i.e., $a_t \sim \pi(\cdot \,|\, s_t)$. The value function $V^\pi : \mathcal{S} \to \mathbb{R}$ and the action-value function (Q-function) $Q^\pi : \mathcal{S} \times \mathcal{A} \to \mathbb{R}$ are defined as

$$V^\pi(s) = \mathbb{E}_\pi \Big[ \sum_{t=0}^\infty \gamma^t r(s_t, a_t) \,\Big|\, s_0 = s \Big], \quad Q^\pi(s, a) = \mathbb{E}_\pi \Big[ \sum_{t=0}^\infty \gamma^t r(s_t, a_t) \,\Big|\, s_0 = s, a_0 = a \Big]. \tag{2}$$

where the expectation is with respect to the trajectory $\tau$ induced by policy $\pi$.

We define the Bellman operator as

$$(\mathbb{B}f)(s, a) = \mathbb{E}_{s' \sim p(\cdot|s,a)} \big[ r(s, a) + \gamma f(s') \big]. \tag{3}$$

We use $\pi^*$, $Q^*$, and $V^*$ to denote the optimal policy, optimal Q-function, and optimal value function, respectively. We have the Bellman optimality equation

$$V^*(s) = \max_{a \in \mathcal{A}} Q^*(s, a), \quad Q^*(s, a) = (\mathbb{B}V^*)(s, a). \tag{4}$$

Meanwhile, the optimal policy $\pi^*$ satisfies

$$\pi^*(\cdot \,|\, s) = \operatorname*{argmax}_\pi \langle Q^*(s, \cdot), \pi(\cdot \,|\, s) \rangle_\mathcal{A}, \quad V^*(s) = \langle Q^*(s, \cdot), \pi^*(\cdot \,|\, s) \rangle_\mathcal{A},$$

where the maximum is taken over all functions mapping from $\mathcal{S}$ to distributions over $\mathcal{A}$. We aim to learn a policy that maximizes the expected cumulative reward. Correspondingly, we define the performance metric as

$$\text{SubOpt}(\pi, s) = V^{\pi^*}(s) - V^\pi(s). \tag{5}$$

### 3.2 PROVABLE OFFLINE ALGORITHMS

In this section, we consider *pessimistic value iteration* (PEVI; Jin et al., 2021) as the backbone algorithm, described in Algorithm 2. It is a representative model-free offline algorithm with theoretical guarantees. PEVI uses negative bonus $\Gamma(\cdot, \cdot)$ over standard $Q$-value estimation $\widehat{Q}(\cdot, \cdot) = (\widehat{\mathbb{B}}\widehat{V})(\cdot)$ to reduce potential bias due to finite data, where $\widehat{\mathbb{B}}$ is some empirical estimation of $\mathbb{B}$ from dataset $\mathcal{D}$. Please refer to Appendix A.1 for more details of the PEVI algorithm.

We use the following notion of $\xi$-uncertainty quantifier as follows to formalize the idea of pessimism.

**Definition 3.2** ($\xi$-Uncertainty Quantifier). We say $\Gamma : \mathcal{S} \times \mathcal{A} \to \mathbb{R}$ is a $\xi$-uncertainty quantifier for $\widehat{\mathbb{B}}$ and $\widehat{V}$ if with probability $1 - \xi$, for all $(s, a) \in \mathcal{S} \times \mathcal{A}$,

$$\big| (\widehat{\mathbb{B}}\widehat{V})(s, a) - (\mathbb{B}\widehat{V})(s, a) \big| \leq \Gamma(s, a). \tag{6}$$

### 3.3 UNSUPERVISED DATA SHARING

We consider unsupervised data sharing in offline reinforcement learning. We first characterize the quality of the dataset with the notion of coverage coefficient (Uehara & Sun, 2021), defined as below.

**Definition 3.3.** The coverage coefficient $C^\dagger$ of a dataset $\mathcal{D} = \{(s_\tau, a_\tau, r_\tau)\}_{\tau=1}^N$ is defined as

$$C^\dagger = \sup_C \left\{ \frac{1}{N} \cdot \sum_{\tau=1}^N \phi(s_\tau, a_\tau)\phi(s_\tau, a_\tau)^\top \succeq C \cdot \mathbb{E}_{\pi^*} \big[ \phi(s_t, a_t)\phi(s_t, a_t)^\top \,|\, s_0 = s \big], \forall s \in \mathcal{S} \right\},$$

The coverage coefficient $C^\dagger$ is common in offline RL literature (Uehara & Sun, 2021; Jin et al., 2021; Rashidinejad et al., 2021), which represents the maximum ratio between the density of empirical state-action distribution and the density induced from the optimal policy. Intuitively, it represents the quality of the dataset. For example, the `expert` dataset has a high coverage ratio while the `random` dataset may have a low ratio.

We denote $\mathcal{D}_0$ as the origin labeled dataset, with coverage coefficient $C_0^\dagger$ and size $N_0$. And we denote the unlabeled dataset as $\mathcal{D}_1$, with coverage coefficient $C_1^\dagger$ and size $N_1$. Note that it is possible that the unlabeled data comes from multiple sources, such as multi-task settings, and we still use $\mathcal{D}_1$ to represent the combined dataset $\cup_{i=1}^M \mathcal{D}_i$ from $M$ tasks for simplicity.

## 4    PROVABLE UNSUPERVISED DATA SHARING

How can we leverage reward-free data for offline RL? A naive approach is to learn the reward function from labeled data via the following regression

$$\widehat{\theta} = \underset{\theta}{\arg\min} \sum_{\tau=1}^{N_0} (f_\theta(s_\tau, a_\tau) - r_\tau)^2 + \frac{\nu}{2}\|\theta\|_2^2, \tag{7}$$

where $f_\theta(s_\tau, a_\tau) = \phi(s_\tau, a_\tau)^\top \theta$ in linear MDPs. Then we can use this learned reward function to label unsupervised data. However, this approach can lead to suboptimal performance due to overestimation of the predicted reward $r_{\widehat{\theta}}$ (Yu et al., 2022), which undermines the pessimism in offline algorithms.

To address this issue, we propose a data-sharing algorithm called Provable Data Sharing (PDS). We start by analyzing the uncertainty in learned reward functions and add penalties for such uncertainty to leverage unlabeled data. In Section 4.2, we show that PDS has a provable performance bound consisting of two parts: the offline error, which is tightened compared to no data sharing due to additional data, and the error from reward bias. The performance bound of PDS is provably better than no data sharing as long as the unlabeled dataset has mediocre size or quality. We also extend our algorithm in linear MDPs to general settings in Section 4.3 and propose using ensembles for reward uncertainty estimation. We demonstrate the effectiveness of PDS by integrating it with IQL (Kostrikov et al., 2021), and present Algorithm 3, which is simple and can be easily integrated with other model-free offline algorithms.

### 4.1    PROVABLE DATA SHARING

To address the issue of potential overestimation of predicted rewards, we first analyze the uncertainty in learned reward functions. In the context of linear MDPs, the reward function can be learned via linear regression, and the uncertainty of the parameters is characterized by the elliptical confidence region, as shown in Lemma 4.1. This confidence region is important as it allows us to give a more accurate estimation of the reward function while keeping the overall algorithm pessimistic.

**Lemma 4.1** ( Abbasi-Yadkori et al. (2011)). *Let* $\alpha = \sqrt{\nu} + r_{\max} \cdot \sqrt{2\log\frac{1}{\delta} + d\log\left(1 + \frac{N_0}{\nu d}\right)}, \Lambda = \nu I + \sum_{\tau=1}^{N_0} \phi(s_\tau, a_\tau)\phi(s_\tau, a_\tau)^\top,$

$$\mathcal{C}(\delta) = \left\{ \theta \in \mathbb{R}^d \mid \|\theta - \widehat{\theta}\|_\Lambda \leq \alpha \right\}, \tag{8}$$

*where* $\widehat{\theta}$ *is the minimizer in Equation* (7)*, then we have* $\mathcal{P}(\theta^\star \in \mathcal{C}(\delta)) \geq 1 - \delta$*, where* $\theta^\star$ *is the true parameter for the reward function.*

*Proof.* Please refer to Theorem 2 in Abbasi-Yadkori et al. (2011) for detailed proof.    □

Lemma 4.1 provides a useful insight: the uncertainty of the learned reward function in linear MDPs only depends on the quality and size of *labeled* data. Based on this insight, we propose a two-phase algorithm that guarantees a provable performance bound, as shown in Theorem 4.3. The simple reward prediction method can compromise the pessimistic estimation of the algorithm, while UDS may result in a reward bias that is too large. Our algorithm consists of two phases: in the first phase,

we construct a pessimistic reward estimator that finds the reward function in the confidence set that leads to the lowest optimal value. In the second phase, we conduct standard offline RL with the given pessimistic reward function.

To solve the challenge of finding the best parameter in the confidence set, which is a bi-level optimization problem, we propose using a simpler method that maintains the pessimistic property of the offline algorithms. This method involves using a pessimistic estimation, which allows us to keep the algorithm pessimistic while avoiding the computational challenges of the bi-level optimization problem. Formally,

$$\widehat{r}(s,a) = \max\left\{\phi(s,a)^{\top}\widehat{\theta} - \alpha\sqrt{\phi(s,a)^{\top}\Lambda^{-1}\phi(s,a)}, 0\right\},\tag{9}$$

where $\Lambda = \nu I + \sum_{\tau=1}^{N_0}\phi(s_\tau,a_\tau)\phi(s_\tau,a_\tau)^{\top}$.

We adopt the pessimistic estimation in Equation (9) because it provides a lower bound for reward functions in the confidence set $\mathcal{C}(\delta)$, as guaranteed by the following lemma derived from Cauchy-Schwartz inequalities.

**Lemma 4.2.** *For any $\theta \in \mathcal{C}(\delta)$,*

$$\left|\phi(s,a)^{\top}\theta - \phi(s,a)^{\top}\widehat{\theta}\right| \leq \alpha\sqrt{\phi(s,a)^{\top}\Lambda^{-1}\phi(s,a)}.\tag{10}$$

When labeled data is scarce or there is a significant distributional shift between the labeled and unlabeled data, Equation (9) degenerates to 0, which is equivalent to the UDS algorithm (Yu et al., 2022).

---

**Algorithm 1** Provable Data Sharing, Linear MDP

1: **Require**: Labeled dataset $\mathcal{D}_0 = \{(s_\tau, a_\tau, r_\tau)\}_{\tau=1}^{N_0}$, unlabeled dataset $\mathcal{D}_1 = \{(s_\tau, a_\tau)\}_{\tau=1}^{N_1}$.
2: **Require**: Confidence parameter $\alpha, \beta, \delta$.
3: Learn the reward function $\widehat{\theta}$ from $\mathcal{D}_0$ using Equation (7)
4: Construct the confidence set $\mathcal{C}(\delta)$ using Equation (8).
5: Construct the pessimistic reward over the confidence set

$$\widetilde{\theta} \leftarrow \underset{\theta \in \mathcal{C}(\delta)}{\arg\min}\, \widehat{V}_\theta,\tag{11}$$

   where $\widehat{V}_\theta$ is the estimated value function from Algorithm 2 and dataset $\mathcal{D}_0 \cup \mathcal{D}_1$ with reward relabeled with parameter $\theta$.
6: Annotate the reward in $\mathcal{D}_0 \cup \mathcal{D}_1$ with parameter $\widetilde{\theta}$.
7: Learn the policy from the annotated dataset $\mathcal{D}_0 \cup \mathcal{D}_1$ using Algorithm 2

$$\widehat{V}, \widehat{\pi} \leftarrow \text{PEVI}(\mathcal{D}_0 \cup \mathcal{D}_1).\tag{12}$$

8: **Return** $\widehat{\pi}$

---

### 4.2 THEORETICAL ANALYSIS

The following subsection analyzes how the provable data-sharing (PDS) algorithm can enhance the performance bound by leveraging unlabeled data. To be specific, we present the following theorem.

**Theorem 4.3** (Performance Bound for PDS). *Suppose the dataset $\mathcal{D}_0, \mathcal{D}_1$ have positive coverage coefficients $C_0^\dagger, C_1^\dagger$, and the underlying MDP is a linear MDP. In Algorithm 1, we set*

$$\lambda = 1, \nu = 1, \alpha = 2\sqrt{d\zeta_2}\cdot r_{\max}, \beta = \frac{cd\sqrt{\zeta_1}}{1-\gamma}\cdot r_{\max}, \zeta_1 = \log\left(\frac{4d(N_0+N_1)}{(1-\gamma)\delta}\right), \zeta_2 = \log\left(\frac{2dN_0}{\delta}\right),$$

*where $c > 0$ is an absolute constant and $\delta \in (0, 1)$ is the confidence parameter. Then with probability $1 - 2\delta$, the policy $\widehat{\pi}$ generated by PDS satisfies for all $s \in \mathcal{S}$,*

$$\text{SubOpt}(\widehat{\pi}; s) \leq \frac{2cr_{\max}}{(1 - \gamma)^2} \sqrt{\frac{d^3 \zeta_1}{N_0 C_0^\dagger + N_1 C_1^\dagger}} + \frac{4r_{\max}}{1 - \gamma} \sqrt{\frac{d^2 \zeta_2}{N_0 C_0^\dagger}}.$$

*Proof.* Please refer to Appendix B for detailed proof. □

The performance bound of PDS is composed of two terms. The first is the offline error, which is inherited from offline algorithms. This bound is improved when additional unlabeled data with size $N_1$ and coverage $C_1^\dagger$ is available. The second term is the reward bias, which arises due to uncertainties in the rewards. Notably, this term is equivalent to the performance bound of a linear bandit with rewards in the range $[0, r_{\max}/(1-\gamma)]$. As the number of unlabeled data approaches infinity, the uncertainty of the dynamics decreases to zero, and the RL problem becomes a linear bandit problem. The theorem demonstrates that PDS outperforms UDS, which suffers from a constant reward bias, and naive reward prediction methods, which lack pessimism and therefore do not offer such guarantees. Moreover, we demonstrate the tightness of the bound by constructing an "adversarial" dataset that matches the bound's suboptimality (see Appendix F).

To better understand the benefits of unlabeled data, we define the suboptimality bound ratio (SBR) of an offline algorithm $\mathcal{A}$ as the ratio of the suboptimality bound obtained by the policy learned with additional unlabeled data to the suboptimality bound of the policy learned using labeled data alone. Mathematically, the SBR of $\mathcal{A}$ is given by:

$$\text{SBR}(\mathcal{A}) = \frac{\overline{\text{SubOpt}}\big(\widehat{\pi}\mathcal{A}(\mathcal{D}_0, \mathcal{D}_1)\big)}{\overline{\text{SubOpt}}\big(\widehat{\pi}\mathcal{A}(\mathcal{D}_0, \varnothing)\big)}, \tag{13}$$

where $\overline{\text{SubOpt}}$ is the tight upper bound on suboptimality. The SBR provides a measure of the benefit of unlabeled data to the offline algorithm, with a smaller SBR indicating a greater benefit from the unlabeled data. Applying this definition to PDS, we obtain the following corollary.

**Corollary 4.4** (Informal). *The SBR of PDS satisfies*

$$\text{SBR} \approx \underbrace{\sqrt{\frac{N_0 C_0^\dagger}{N_0 C_0^\dagger + N_1 C_1^\dagger}}}_{\text{finite sample term}} + \underbrace{\frac{2(1 - \gamma)}{c\sqrt{d}}}_{\text{asymptotic term}}, \tag{14}$$

*where $c$ is the constant in Theorem 4.3 and we ignore the logarithmic factors.*

**When does unlabeled data improve the performance of offline algorithms?** Corollary 4.4 allows us to analyze the relative performance of PDS under different conditions. The first term of the bound depends on the qualities and amounts of both labeled and unlabeled datasets. If the unlabeled dataset has a mediocre number of samples or data quality, the first term will be sufficiently small. The second term affects the asymptotic performance when the unlabeled data approaches infinity, and it depends on the discount factor and the dimension of the problem. PDS improves over no data-sharing algorithms asymptotically in larger problems or longer horizons. For a more detailed discussion, please refer to Appendix E.

## 4.3 PRACTICAL IMPLEMENTATION

This subsection outlines the practical implementation of PDS in general MDPs. We employ $L$ ensembles $\theta_1, \ldots, \theta_L$ to estimate uncertainty, which are learned using Equation (7). To estimate pessimistic rewards, we use the following pessimistic estimation:

$$\widehat{r}(s, a) = \max\{\mu(s, a) - k\sigma(s, a), 0\}, \tag{15}$$

where $\mu(s, a) = \frac{1}{L} \sum_{i=1}^{L} f_{\theta_i}(s, a), \sigma(s, a) = \sqrt{\frac{1}{L} \sum_{i=1}^{L} (f_{\theta_i}(s, a) - \mu(s, a))^2}$ are the mean and standard deviation, respectively. Here, $k$ is a hyperparameter used to control the amount of pessimism. We can also use the minimum over $L$ ensembles for the pessimistic estimation, which is

linked to Equation (15) following An et al. (2021); Royston (1982) as shown in Equation (16):

$$\mathbb{E}\left[\min_{j=1,\ldots,L} f_{\theta_j}(s,a)\right] \approx \mu(s,a) - \Phi^{-1}\left(\frac{L - \frac{\pi}{8}}{L - \frac{\pi}{4} + 1}\right)\sigma(s,a), \qquad (16)$$

where $\Phi$ is the CDF of the standard Gaussian distribution.

The appropriate value of $k$ for each domain can be difficult to determine. To address this issue, we observe that the amount of pessimism required for different domains is proportional to the difference in mean rewards between labeled and unlabeled data. Leveraging this insight, we propose a simple and efficient automatic mechanism for adjusting the value of the $k$ parameter. Specifically, we suggest a method that adjusts $k$ based on the difference in mean rewards, as given by Equation (17):

$$\widehat{r}(s,a) = \max\left\{\min_{j=1,\ldots,L} f_{\theta_j}(s,a) - k\sigma(s,a), 0\right\},$$
$$\text{where} \quad k = a \cdot \frac{\max(\mu - \widehat{\mu}, 0)}{|\mu| + \epsilon}. \qquad (17)$$

where $\mu = \frac{1}{N_0}\sum_{i=1}^{N_0} \mu(s_i, a_i), \widehat{\mu} = \frac{1}{N_1}\sum_{i=1}^{N_1} \widehat{\mu}(s_i, a_i)$ are the mean reward of labeled and (predicted) unlabeled data, respectively. We use $a = 25$ and $L = 10$ in all experiments.

Then we can plug in any model-free offline algorithms. Here we use IQL (Kostrikov et al., 2021) as the backbone offline algorithm, but we emphasize that it can be easily integrated with other model-free algorithms. The details of our algorithm is summarized in Algorithm 3 in Appendix A.2.

## 5 EXPERIMENTS

In this section, we aim to evaluate the effectiveness of pessimistic reward estimation and answer the following questions: (1) How does PDS perform compared to the naive reward prediction and unlabeled data sharing (UDS) methods in single locomotion and manipulation tasks? (2) How does PDS behave in multi-task offline RL settings compared to baselines? (3) What makes PDS effective?

| Algorithm | door-open | door-close | drawer-open | drawer-close | average |
|-----------|-----------|------------|-------------|--------------|---------|
| UDS | 16.2±12.1 | 0.0±0.0 | 30.4±60.4 | **182.2±0.4** | 57.2±30.8 |
| Rew Pred | **26.4±12.9** | 110.8±14.3 | 102.6±40.2 | **182.2±0.4** | 105.5±22.3 |
| PDS | **25.5±15.5** | **114.3±1.8** | **153.8±0.4** | **182.8±0.4** | **119.1±18.1** |
| No Share | 4.8±9.5 | 0.0±0.0 | 29.6±58.7 | 175.0±17.6 | 52.4±31.0 |
| Oracle | 20.6±13.3 | 113.2±5.7 | 135.6±36.5 | 182.6±0.4 | 113.0±19.6 |

Table 1: Experiment results for multi-task robotic manipulation (Meta-World) experiments. Numbers are averaged across five seeds and we bold the best-performing method that does not have access to the true rewards.

| Env | Tasks / Dataset type | UDS | Rew Pred | CDS+UDS | PDS |
|-----|---------------------|-----|----------|---------|-----|
| Antmaze | medium-play (3 tasks) / directed | 15.8±1.2 | 26.2±3.7 | **40.6±4.0** | **40.0±3.6** |
| | medium-play (3 tasks) / undirected | 19.6±2.5 | 27.2±2.9 | **37.2±5.1** | 29.2±4.1 |
| | medium-diverse (3 tasks) / directed | 8.7±3.3 | 20.2±3.8 | 33.3±11.5 | **53.2±3.6** |
| | medium-diverse (3 tasks) / undirected | 9.7±1.3 | **42.2±4.3** | 40.7±3.5 | **42.5±5.2** |

Table 2: Experiment results for AntMaze tasks with normalized score metric averaged with five random seeds.

**Single-task domains and datasets.** To address Question (1), we empirically evaluate the PDS algorithm on the hopper, walker2d, and kitchen tasks from the D4RL benchmark suite (Fu et al., 2020). We use 50 labeled trajectories with varying amounts of unlabeled data of different sizes and qualities. This experimental setup is motivated by real-world problems where labeled data is often scarce, and additional unlabeled data may be readily available.

| Task | Labeled Size | Unlabeled Size | UDS | Rew Pred | PDS | Oracle |
|------|--------------|----------------|-----|----------|-----|--------|
| Hopper | medium / 50K | medium / 0.1M | 58.7±1.5 | 64.8±3.2 | **73.9±8.4** | 66.3±2.1 |
| | medium / 50K | medium / 0.4M | 57.3±1.4 | 68.9±4.0 | **77.8±7.4** | 67.4±4.2 |
| | medium / 50K | medium / 0.6M | 56.6±1.4 | 68.5±2.1 | **75.9±2.4** | 68.2±2.1 |
| | expert / 50K | random / 0.1M | **53.3±3.8** | 27.9±15.1 | 42.7±9.8 | 18.1±5.9 |
| | random / 50K | expert / 0.1M | 4.3±0.4 | 84.7±10.8 | **92.3±9.8** | 40.1±4.5 |
| Walker2d | medium / 50K | medium / 0.1M | 70.8±1.2 | 71.4±2.9 | **76.1±0.2** | 74.6±2.3 |
| | medium / 50K | medium / 0.4M | 75.3±1.4 | 70.9±4.1 | **80.1±0.3** | 77.7±2.4 |
| | medium / 50K | medium / 0.6M | 74.8±0.4 | **79.9±4.2** | 79.1±1.4 | 79.1±2.7 |
| | expert / 50K | random / 0.1M | 25.7±13.1 | 2.7±0.2 | **39.5±10.0** | 22.6±1.2 |
| | random / 50K | expert / 0.1M | 0.4±0.1 | 95.3±2.3 | **101.4±3.2** | 15.8±1.3 |

Table 3: Experimental results with normalized score metric averaged with five random seeds.

**Multi-task domains and datasets.** We investigate Question (2) by evaluating PDS on several multi-task domains. The first set of domains we consider is Meta-World (Yu et al., 2020a), where we adopt the same setup as in CDS (Yu et al., 2021a) and evaluate PDS on four tasks: `door open`, `door close`, `drawer open`, and `drawer close`. The second domain is the AntMaze task in D4RL, which consists of mazes of two sizes (medium and large) and includes 3 and 7 tasks, respectively, corresponding to different goal positions. For a detailed description of the experimental setting, please refer to Appendix D.

**Comparisons.** To ensure a fair comparison, we combine UDS with IQL (Kostrikov et al., 2021), the same underlying offline RL method as PDS. In addition to UDS, we train a naive reward prediction method and the sharing-all-true-rewards method (Oracle), and adapt them with IQL. In all experiments, we set the hyperparameters $a = 25$ and $L = 10$ for our method.

## 5.1 Experimental Results

**Results of Question (1).** We evaluated each method on the hopper, walker2d, and kitchen domains and found that PDS outperformed the other methods on most tasks and achieved competitive or better performance than the oracle method (Table 3). Notably, PDS performed well when the labeled and unlabeled datasets had different data qualities, which we attribute to its ability to capture the uncertainties induced by this distribution shift and maintain a pessimistic algorithm. The prediction method performed well when the unlabeled dataset had high quality, and UDS performed well when the unlabeled data had low quality. PDS combined the strengths of both methods and achieved superior performance.

**Results of Question (2).** Multi-task settings exhibit greater distributional shifts between labeled and unlabeled data due to the differing task goals. We evaluated PDS and the other methods on the AntMaze and Meta-World domains (Tables 2 and 1) and found that PDS's performance was comparable to the oracle method and outperformed the other methods. UDS performed relatively poorly on the Meta-World dataset, possibly due to the high dataset quality, which made labeling with zeros induce a large reward bias. On the multi-task AntMaze domain, PDS outperformed both UDS and the naive prediction method, especially on the diverse dataset. These results aligned with our observation on single-task domains that PDS performs better when the distribution shift between datasets is larger.

**Results of Question (3).** We conducted experiments on the hopper and walker2d tasks with various penalty weights $k$ to investigate the effect of uncertainty weights in PDS (Figure 1). The results shows that PDS can interpolate between UDS and the reward prediction method and offered a better trade-off to balance the conservation and generalization of reward estimators, resulting in better performance. Also, PDS reduces the variance from reward prediction and is close to the oracle method, indicating its ability to reduce the uncertainties from the variance of reward predictors while keeping the reward bias small, as shown in Figure 2.

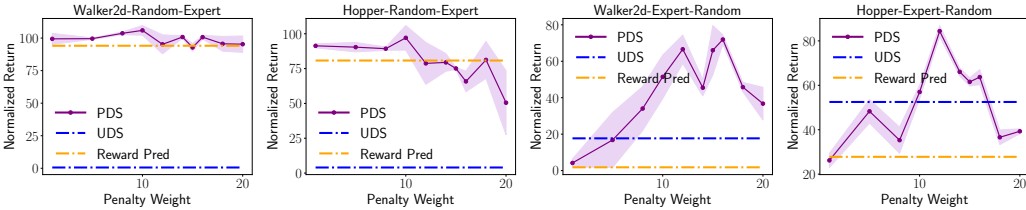

Figure 1: Impact of penalty weight $k$ on the performance. We evaluate PDS on Hopper/Walker2d-Labeled (Expert/Random)-Unlabeled (Expert/Random) tasks with various $k$.

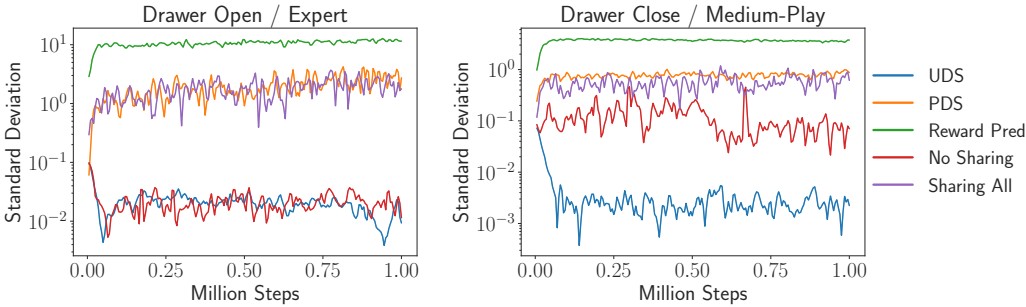

Figure 2: The standard deviation of the state values learned with IQL. The reward prediction method suffers from a large deviation. The oracle method's and PDS's deviation is mediocre, and no sharing and UDS methods enjoy the smallest deviation.

**Discussion of PDS, UDS, and Reward Prediction:** PDS can be seen as a generalization of both UDS and the reward prediction method. UDS sets the penalty weight $k$ to infinity, while the reward prediction method sets it to zero. However, UDS introduces a high reward bias, and the reward prediction method ruins the pessimism of offline algorithms. In contrast, PDS offers a trade-off between bias and pessimism by adaptively adjusting $k$. To verify that overestimation is the main factor for the suboptimal performance of the reward prediction method, we conduct additional ablation studies, as shown in Appendix G.

## 6 CONCLUSION

In this paper, we show that incorporating reward-free data into offline reinforcement learning can yield significant performance improvements. Our theoretical analysis reveals that unlabeled data provides additional information about the MDP's dynamics, reducing the problem to linear bandits in the limit and improving performance bounds therefore. Building upon these insights, we propose a new algorithm, PDS, that leverages this information by incorporating uncertainty penalties on learned rewards to ensure a conservative approach. Our method has provable guarantees in theory and achieves superior performance in practice. In future work, it may be interesting to explore how PDS can be further improved with representation learning methods, and to extend our analysis to more general settings, such as generalized linear MDPs (Wang et al., 2019) and low-rank MDPs (Ayoub et al., 2020; Jiang et al., 2017).

## 7 ACKNOWLEDGEMENTS

This work is supported in part by Science and Technology Innovation 2030 - "New Generation Artificial Intelligence" Major Project (No. 2018AAA0100904) and the National Natural Science Foundation of China (62176135).

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

# A ALGORITHM DETAILS

## A.1 PESSIMISTIC VALUE ITERATION (PEVI,(JIN ET AL., 2021))

In this section, we describe the details of PEVI algorithm.

In linear MDPs, we can construct $\widehat{\mathbb{B}}\widehat{V}$ and $\Gamma$ based on $\mathcal{D}$ as follows, where $\widehat{\mathbb{B}}\widehat{V}$ is the empirical estimation for $\mathbb{B}\widehat{V}$. For a given dataset $\mathcal{D} = \{(s_\tau, a_\tau, r_\tau)\}_{\tau=1}^N$, we define the empirical mean squared Bellman error (MSBE) as

$$M(w) = \sum_{\tau=1}^N \big(r_\tau + \gamma\widehat{V}(s_{\tau+1}) - \phi(s_\tau, a_\tau)^\top w\big)^2 + \lambda||w||_2^2$$

Here $\lambda > 0$ is the regularization parameter. Note that $\widehat{w}$ has the closed form

$$\widehat{w} = \Lambda^{-1}\Big(\sum_{\tau=1}^N \phi(s_\tau, a_\tau) \cdot \big(r_\tau + \gamma\widehat{V}(s_{\tau+1})\big)\Big),$$

$$\text{where } \Lambda = \lambda I + \sum_{\tau=1}^N \phi(s_\tau, a_\tau)\phi(s_\tau, a_\tau)^\top. \tag{18}$$

Then we simply let

$$\widehat{\mathbb{B}}\widehat{V} = \langle\phi, \widehat{w}\rangle. \tag{19}$$

Meanwhile, we construct $\Gamma$ based on $\mathcal{D}$ as

$$\Gamma(s, a) = \beta \cdot \big(\phi(s, a)^\top\Lambda^{-1}\phi(s, a)\big)^{1/2}. \tag{20}$$

Here $\beta > 0$ is the scaling parameter. The overall PEVI algorithm is summarized in Algorithm 2.

---

**Algorithm 2** Pessimistic Value Iteration, PEVI

1: **Require**: Dataset $\mathcal{D} = \{(s_\tau, a_\tau, r_\tau, s_{\tau+1})\}_{\tau=1}^T$.
2: Initialization: Set $\widehat{V}(\cdot) \leftarrow 0$ and construct $\Gamma(\cdot, \cdot)$.
3: **while** not converged **do**
4:     Construct $(\widehat{\mathbb{B}}\widehat{V})(\cdot, \cdot)$
5:     Set $\widehat{Q}(\cdot, \cdot) \leftarrow (\widehat{\mathbb{B}}\widehat{V})(\cdot, \cdot) - \Gamma(\cdot, \cdot)$.
6:     Set $\widehat{\pi}(\cdot\,|\,\cdot) \leftarrow \text{argmax}_\pi \mathbb{E}_\pi\Big[\widehat{Q}(\cdot, \cdot)\Big]$.
7:     Set $\widehat{V}(\cdot) \leftarrow \mathbb{E}_{\widehat{\pi}}\Big[\widehat{Q}(\cdot, \cdot)\Big]$.
8: **end while**
9: **Return** $\widehat{V}, \widehat{\pi}$

---

## A.2 IQL WITH PROVABLE DATA SHARING

In this section, we give a detailed description of our IQL+PDS algorithm.

---

**Algorithm 3** IQL+PDS algorithm, General MDPs

---

**Input**: Labeled dataset $\mathcal{D}_0$, unlabeled dataset $\mathcal{D}_1$.
**Input**: Parameter $\alpha$, $\beta$, $k$, $\tau$.
**Output**: policy $\pi_\phi$.

 1: Learn $L$ reward functions as in Equation (7).
 2: Construct pessimistic reward estimation as in Equation (17).
 3: Relabel unsupervised dataset $\mathcal{D}_1$ and combine with the labeled dataset $\mathcal{D}_0$.
 4: Initialize $\psi$, $\theta$, $\widehat{\theta}$, $\phi$.
 5: **for** each gradient step **do**
 6:     $\phi \leftarrow \psi - \lambda_V \nabla_\psi L_V(\psi)$,    $L_V(\psi) = \mathbb{E}_{s,a}\left[L_2^\tau(Q_{\widehat{\theta}}(s,a) - V_\psi(s))\right]$
 7:     $\theta \leftarrow \theta - \lambda_Q \nabla_\theta L_Q(\theta)$,    $L_Q(\theta) = \mathbb{E}_{s,a,r}\left[(r + \gamma Q_{\widehat{\theta}}(s,a) - Q_\theta(s,a))^2\right]$
 8:     $\widehat{\theta} \leftarrow \alpha\theta + (1-\alpha)\theta$
 9: **end for**
10: **for** each gradient step **do**
11:     $\phi \leftarrow \phi - \lambda_\pi \nabla_\phi L_\pi(\phi)$,    $L_\pi(\phi) = \mathbb{E}_{s,a}\left[\exp\left(\beta(Q_{\widehat{\theta}}(s,a) - V_\psi(s))\right)\log(\pi_\phi(a|s))\right]$
12: **end for**

---

## B   PROOF OF THEOREM 4.3

*Proof.* From Equation (11) in Algorithm 1, we have

$$\widehat{V}_{\widetilde{\theta}} \leq \widehat{V}_\theta, \quad \forall \theta \in \mathcal{C}(\delta), \tag{21}$$

where $\widetilde{\theta}$ is the pessimistic estimation of $\theta$.

Let $\mathcal{E}_1$ be the event $\theta^\star \in \mathcal{C}(\delta)$, then we have $\mathcal{P}(\mathcal{E}_1) \geq 1 - \delta$ from Lemma 4.1.

Let $\mathcal{E}_2$ be the event where the following inequality holds,

$$|(\mathbb{B}\widehat{V})(s,a) - (\widehat{\mathbb{B}}\widehat{V})(s,a)| \leq \Gamma = \beta\sqrt{\phi(s,a)^\top \Lambda^{-1}\phi(s,a)}, \forall(s,a) \in \mathcal{S} \times \mathcal{A}. \tag{22}$$

then we have $\mathcal{P}(\mathcal{E}_2) \geq 1 - \delta$ from Lemma C.3.

Condition on $\mathcal{E}_1 \cap \mathcal{E}_2$, we have.

$$
\begin{aligned}
V_{\theta^\star}^{\pi^*} - V_{\theta^\star}^{\widehat{\pi}} &= V_{\theta^\star}^{\pi^*} - \widehat{V}_{\theta^\star} + \widehat{V}_{\theta^\star} - V_{\theta^\star}^{\widehat{\pi}} \\
&\leq V_{\theta^\star}^{\pi^*} - \widehat{V}_{\theta^\star} \\
&= V_{\theta^\star}^{\pi^*} - V_{\widehat{\theta}}^{\pi^*} + V_{\widehat{\theta}}^{\pi^*} - \widehat{V}_{\widetilde{\theta}} + \widehat{V}_{\widetilde{\theta}} - \widehat{V}_{\theta^\star} \\
&\leq V_{\theta^\star}^{\pi^*} - V_{\widehat{\theta}}^{\pi^*} + V_{\widehat{\theta}}^{\pi^*} - \widehat{V}_{\widetilde{\theta}} \\
&= V_{\theta^\star}^{\pi^*} - V_{\widehat{\theta}}^{\pi^*} + V_{\widehat{\theta}}^{\pi^*} - V_{\widetilde{\theta}}^{\pi^*} + V_{\widetilde{\theta}}^{\pi^*} - \widehat{V}_{\widetilde{\theta}} \\
&\leq \frac{4r_{\max}}{1-\gamma}\sqrt{\frac{d^2\zeta_2}{N_0 C_0^\dagger}} + \frac{2cr_{\max}}{(1-\gamma)^2}\sqrt{\frac{d^3\zeta_1}{N_0 C_0^\dagger + N_1 C_1^\dagger}},
\end{aligned}
$$

where the first inequality follows from Lemma C.2. The second inequality follows from Equation (21), and the last inequality follows from Lemma C.5 and C.1.

From the union bound, we have that the above inequality holds with a probability of $1 - 2\delta$.    $\square$

## C   ADDTIONAL LEMMAS AND MISSING PROOFS

**Lemma C.1.** *Under the event in Lemma C.3, we have*

$$V_\theta^{\pi^*}(s) - \widehat{V}_\theta(s) \leq \frac{2cr_{max}}{(1-\gamma)^2}\sqrt{\frac{d^3\zeta}{C^\dagger N}},$$

*with probability* $1 - \delta$, *for all* $\|\theta\|_2^2 \le d$.

*Proof.* Let

$$\delta(s, a) = r(s, a) + \gamma \mathbb{E}_{s' \sim \mathcal{P}(\cdot|s,a)} \widehat{V}(s') - \widehat{Q}(s, a), \tag{23}$$

From the definition of $\widehat{Q}(s, a)$ and $\widehat{V}(s)$, we have

$$\delta(s, a) = \mathbb{B}_\gamma \widehat{V}(s) - \widehat{Q}(s, a) = \mathbb{B}_\gamma \widehat{V}(s) - \widehat{\mathbb{B}}_\gamma \widehat{V} + \Gamma(s, a). \tag{24}$$

Under the condition of Lemma C.3, it holds that

$$0 \le \delta(s, a) \le 2\Gamma(s, a), \text{ for all } s, a. \tag{25}$$

Then we have

$$
\begin{aligned}
& V_\theta^{\pi^*}(s) - \widehat{V}_\theta(s) \\
=& \mathbb{E}_{a \sim \pi^*, s' \sim \mathcal{P}(\cdot|s,a)} \left[ r(s, a) + \gamma V^{\pi^*}(s') \right] - \mathbb{E}_{a \sim \widehat{\pi}} \left[ \widehat{Q}(s, a) \right] \\
=& \mathbb{E}_{a \sim \pi^*, s' \sim \mathcal{P}(\cdot|s,a)} \left[ r(s, a) + \gamma V^{\pi^*}(s') - \widehat{Q}(s, a) \right] + \mathbb{E}_{a \sim \pi^*} \left[ \widehat{Q}(s, a) \right] - \mathbb{E}_{a \sim \widehat{\pi}} \left[ \widehat{Q}(s, a) \right] \\
=& \mathbb{E}_{a \sim \pi^*, s' \sim \mathcal{P}(\cdot|s,a)} \left[ r(s, a) + \gamma \widehat{V}(s') - \widehat{Q}(s, a) \right] + \gamma \mathbb{E}_{a \sim \pi^*, s' \sim \mathcal{P}(\cdot|s,a)} \left[ V^{\pi^*}(s') - \widehat{V}(s') \right] \\
& + \left\langle \widehat{Q}(s, \cdot), \pi^*(\cdot \,|\, s) - \widehat{\pi}(\cdot \,|\, s) \right\rangle_{\mathcal{A}} \\
=& \mathbb{E}_{a \sim \pi^*, s' \sim \mathcal{P}(\cdot|s,a)} \left[ \delta(s, a) \right] + \left\langle \widehat{Q}(s, \cdot), \pi^*(\cdot \,|\, s) - \widehat{\pi}(\cdot \,|\, s) \right\rangle_{\mathcal{A}} + \cdots \\
=& \mathbb{E}_{\pi^*} \left[ \sum_{t=0}^{\infty} \gamma^t \delta(s_t, a_t) \,|\, s_0 = s \right] + \mathbb{E}_{\pi^*} \left[ \sum_{t=0}^{\infty} \gamma^t \left\langle \widehat{Q}(s_t, \cdot), \pi^*(\cdot \,|\, s_t) - \widehat{\pi}(\cdot \,|\, s_t) \right\rangle_{\mathcal{A}} \,|\, s_0 = s \right] \\
\le& \mathbb{E}_{\pi^*} \left[ \sum_{t=0}^{\infty} \gamma^t \delta(s_t, a_t) \,\Big|\, s_0 = s \right] \\
\le& 2 \mathbb{E}_{\pi^*} \left[ \sum_{t=0}^{\infty} \gamma^t \Gamma(s_t, a_t) \,\Big|\, s_0 = s \right] \\
=& 2\beta \mathbb{E}_{\pi^*} \left[ \sum_{t=0}^{\infty} \gamma^t \left( \phi(s_t, a_t)^\top \Lambda^{-1} \phi(s_t, a_t) \right)^{1/2} \,\Big|\, s_0 = s \right].
\end{aligned}
$$

Here the first inequality follows from the fact that $\widehat{\pi}(\cdot|s) = \operatorname{argmax}_\pi \left\langle \widehat{Q}(\cdot, \cdot), \pi(\cdot|s) \right\rangle$ and the second inequality follows from Equation (25).

By the Cauchy-Schwarz inequality, we have

$$
\begin{aligned}
& \mathbb{E}_{\pi^*} \left[ \sum_{t=0}^{\infty} \gamma^t \left( \phi(s_t, a_t)^\top \Lambda^{-1} \phi(s_t, a_t) \right)^{1/2} \,\Big|\, s_0 = s \right] \\
=& \frac{1}{1-\gamma} \mathbb{E}_{d^{\pi^*}} \left[ \sqrt{\operatorname{Tr} \left( \phi(s, a)^\top \Lambda^{-1} \phi(s, a) \right)} \,\Big|\, s_0 = s \right] \\
=& \frac{1}{1-\gamma} \mathbb{E}_{d^{\pi^*}} \left[ \sqrt{\operatorname{Tr} \left( \phi(s, a) \phi(s, a)^\top \Lambda^{-1} \right)} \,\Big|\, s_0 = s \right] \\
\le& \frac{1}{1-\gamma} \sqrt{\operatorname{Tr} \left( \mathbb{E}_{d^{\pi^*}} \left[ \phi(s, a) \phi(s, a)^\top \,|\, s_0 = s \right] \Lambda^{-1} \right)} \\
=& \frac{1}{1-\gamma} \sqrt{\operatorname{Tr} \left( \Sigma_{\pi^*, s}^\top \Lambda^{-1} \right)}, \tag{26}
\end{aligned}
$$

for all $s \in \mathcal{S}$. Then we have

$$
\begin{aligned}
V_\theta^{\pi^*}(s) - \widehat{V}_\theta(s) &\le 2\beta \mathbb{E}_{\pi^*}\left[\sum_{t=0}^{\infty} \gamma^t \big(\phi(s_t, a_t)^\top \Lambda^{-1} \phi(s_t, a_t)\big)^{1/2} \,\Big|\, s_0 = s\right] \\
&\le \frac{2\beta}{1-\gamma} \sqrt{\operatorname{Tr}\left(\Sigma_{\pi^*,s} \cdot \big(I + C^\dagger \cdot N \cdot \Sigma_{\pi^*,s}\big)^{-1}\right)} \\
&= \frac{2\beta}{1-\gamma} \sqrt{\sum_{j=1}^{d} \frac{\lambda_j(s)}{1 + C^\dagger \cdot N \cdot \lambda_j(s)}}.
\end{aligned}
\tag{27}
$$

Here $\{\lambda_j(s)\}_{j=1}^{d}$ are the eigenvalues of $\Sigma_{\pi^*,s}$ for all $s \in \mathcal{S}$, the second inequality follows from Equation (26). Meanwhile, by Definition 3.1, we have $\|\phi(s,a)\| \le 1$ for all $(s,a) \in \mathcal{S} \times \mathcal{A}$. By Jensen's inequality, we have

$$
\|\Sigma_{\pi^*,s}\|_{\mathrm{op}} \le \mathbb{E}_{\pi^*}\left[\|\phi(s,a)\phi(s,a)^\top\|_{\mathrm{op}} \,\big|\, s_0 = s\right] \le 1
\tag{28}
$$

for all $s \in \mathcal{S}$. As $\Sigma_{\pi^*,s}$ is positive semidefinite, we have $\lambda_j(s) \in [0,1]$ for all $s \in \mathcal{S}$ and all $j \in [d]$. Hence we have

$$
\begin{aligned}
V_\theta^{\pi^*}(s) - \widehat{V}_\theta(s) &\le \frac{2\beta}{1-\gamma} \sqrt{\sum_{j=1}^{d} \frac{\lambda_j(s)}{1 + C^\dagger \cdot N \cdot \lambda_j(s)}} \\
&\le \frac{2\beta}{1-\gamma} \sqrt{\sum_{j=1}^{d} \frac{1}{1 + C^\dagger \cdot N}} \le \frac{2c r_{\max}}{(1-\gamma)^2} \sqrt{\frac{d^3 \zeta}{C^\dagger N}}
\end{aligned}
\tag{29}
$$

for all $x \in \mathcal{S}$, where the second inequality follows from the fact that $\lambda_j(s) \in [0,1]$ for all $s \in \mathcal{S}$ and all $j \in [d]$, while the third inequality follows from the choice of the scaling parameter $\beta > 0$.

Then we have the conclusion in Lemma C.1. $\qquad \square$

**Lemma C.2.** *Under the event in Lemma C.3, we have*

$$
\widehat{V}_\theta(s) - V^{\widehat{\pi}_\theta}(s) \le 0
\tag{30}
$$

*with probability $1 - \delta$, for all $\|\theta\|_2^2 \le d$.*

*Proof.* Similar to the proof of Lemma C.1, let

$$
\delta(s,a) = r(s,a) + \gamma \mathbb{E}_{s' \sim \mathcal{P}(\cdot|s,a)} \widehat{V}(s') - \widehat{Q}(s,a),
\tag{31}
$$

we have

$$
\begin{aligned}
\widehat{V}(s) - V^{\widehat{\pi}}(s) &= \mathbb{E}_{a \sim \widehat{\pi}}\left[\widehat{Q}(s,a)\right] - \mathbb{E}_{a \sim \widehat{\pi}, s' \sim \mathcal{P}(\cdot|s,a)}\left[r(s,a) + \gamma V^{\widehat{\pi}}(s')\right] \\
&= \mathbb{E}_{a \sim \widehat{\pi}, s' \sim \mathcal{P}(\cdot|s,a)}\left[\widehat{Q}(s,a) - r(s,a) - \gamma \widehat{V}(s')\right] \\
&\quad + \gamma \mathbb{E}_{a \sim \widehat{\pi}, s' \sim \mathcal{P}(\cdot|s,a)}\left[\widehat{V}(s') - V^{\widehat{\pi}}(s')\right] \\
&= -\mathbb{E}_{\widehat{\pi}}\left[\delta(s,a)\right] + \gamma \mathbb{E}_{a \sim \widehat{\pi}, s' \sim \mathcal{P}(\cdot|s,a)}\left[\widehat{V}(s') - V^{\widehat{\pi}}(s')\right] \\
&= -\mathbb{E}_{\widehat{\pi}}\left[\delta(s,a)\right] + \cdots \\
&= -\mathbb{E}_{\widehat{\pi}}\left[\sum_{t=0}^{\infty} \gamma^t \delta(s_t, a_t) \,\big|\, s_0 = s\right].
\end{aligned}
$$

Then under the condition of Lemma C.3, it holds that

$$
0 \le \delta(s,a) \le 2\Gamma(s,a), \text{ for all } s, a,
\tag{32}
$$

Then we have the result immediately.

$\qquad \square$

**Lemma C.3** ($\xi$-Quantifiers). *Let*

$$\lambda = 1, \quad \beta = c \cdot dV_{max}\sqrt{\zeta}, \quad \zeta = \log\left(2dN/(1-\gamma)\xi\right). \tag{33}$$

*Then* $\Gamma = \beta \cdot \left(\phi(s,a)^\top \Lambda^{-1}\phi(s,a)\right)^{1/2}$ *are $\xi$-quantifiers with probability at least $1 - \xi$. That is, let $\mathcal{E}_2$ be the event that the following inequality holds,*

$$|(\mathbb{B}\widehat{V})(s,a) - (\widehat{\mathbb{B}}\widehat{V})(s,a)| \leq \Gamma = \beta\sqrt{\phi(s,a)^\top\Lambda^{-1}\phi(s,a)}, \forall(s,a) \in \mathcal{S}\times\mathcal{A}. \tag{34}$$

*Then we have* $\mathcal{P}(\mathcal{E}_2) \geq 1 - \varepsilon$.

*Proof.* we have

$$\mathbb{B}\widehat{V} - \widehat{\mathbb{B}}\widehat{V} = \phi(s,a)^\top(w - \widehat{w})$$

$$= \phi(s,a)^\top w - \phi(s,a)\Lambda^{-1}\left(\sum_{\tau=1}^{N}\phi_\tau(r_\tau + \gamma\widehat{V}(s_{\tau+1}))\right)$$

$$= \underbrace{\phi(s,a)^\top w - \phi(s,a)\Lambda^{-1}\left(\sum_{\tau=1}^{N}\phi_\tau\phi_\tau^\top w\right)}_{\text{(i)}} + \underbrace{\phi(s,a)\Lambda^{-1}(\sum_{\tau=1}^{N}\phi_\tau\phi_\tau^\top w - \sum_{\tau=1}^{N}\phi_\tau(r_\tau + \gamma\widehat{V}(s_{\tau+1})))}_{\text{(ii)}}, \tag{35}$$

Then we bound (i) and (ii), respectively.

For (i), we have

$$\text{(i)} = \phi(s,a)^\top w - \phi(s,a)\Lambda^{-1}(\Lambda - \lambda I)w$$
$$= \lambda\phi(s,a)\Lambda^{-1}w$$
$$\leq \lambda||\phi(s,a)||_{\lambda^{-1}}||w||_{\lambda^{-1}}$$
$$\leq V_{\max}\sqrt{d\lambda}\sqrt{\phi(s,a)^\top\Lambda^{-1}\phi(s,a)}, \tag{36}$$

where the first inequality follows from Cauchy-Schwartz inequality. The second inequality follows from the fact that $||\Lambda^{-1}||_{\text{op}} \leq \lambda^{-1}$ and Lemma C.4.

For notation simplicity, let $\epsilon_\tau = r_\tau + \gamma\widehat{V}(s_{\tau+1}) - \phi_\tau^\top w$, then we have

$$|\text{(ii)}| = \phi(s,a)\Lambda^{-1}\sum_{\tau=1}^{N}\phi_\tau\epsilon_\tau$$

$$\leq ||\sum_{\tau=1}^{N}\phi_\tau\epsilon_\tau||_{\Lambda^{-1}} \cdot ||\phi(s,a)||_{\Lambda^{-1}}$$

$$= \underbrace{||\sum_{\tau=1}^{N}\phi_\tau\epsilon_\tau||_{\Lambda^{-1}}}_{\text{(iii)}} \cdot \sqrt{\phi(s,a)^\top\Lambda^{-1}\phi(s,a)}. \tag{37}$$

The term (iii) is depend on the randomness of the data collection process of $\mathcal{D}$. To bound this term, we resort to uniform concentration inequalities to upper bound

$$\sup_{V\in\mathcal{V}(R,B,\lambda)}\left\|\sum_{\tau=1}^{N}\phi(s_\tau,a_\tau)\cdot\epsilon_\tau(V)\right\|,$$

where

$$\mathcal{V}(R,B,\lambda) = \{V(s;w,\beta,\Sigma) : \mathcal{S}\to[0,V_{\max}] \text{ with}||w|| \leq R, \beta\in[0,B], \Sigma\succeq\lambda\cdot I\}, \tag{38}$$

where $V(s;w,\beta,\Sigma) = \max_a\{\phi(s,a)^\top w - \beta\cdot\sqrt{\phi(s,a)^\top\Sigma^{-1}\phi(s,a)}\}$. For all $\epsilon > 0$, let $\mathcal{N}(\epsilon;R,B,\lambda)$ be the minimal cover if $\mathcal{V}(R,B,\lambda)$. That is, for any function $V \in \mathcal{V}(R,B,\lambda)$, there exists a function $V^\dagger \in \mathcal{N}(\epsilon;R,B,\lambda)$, such that

$$\sup_{s\in\mathcal{S}}|V(s) - V^\dagger(s)| \leq \epsilon. \tag{39}$$

Let $R_0 = V_{\max}\sqrt{Nd/\lambda}, B_0 = 2\beta$, it is easy to show that at each iteration, $\widehat{V}^u \in \mathcal{V}(R_0, B_0, \lambda)$. From the definition of $\mathbb{B}$, we have

$$|\mathbb{B}\widehat{V} - \mathbb{B}V^\dagger| = \gamma \left| \int (\widehat{V}(s') - V^\dagger(s')) \langle \phi(s,a), \mu(s') \rangle \, \mathrm{d}s' \right| \leq \gamma\epsilon. \tag{40}$$

Then we have

$$|(r + \gamma V - \mathbb{B}V) - (r + \gamma V^\dagger - \mathbb{B}V^\dagger)| \leq 2\gamma\epsilon. \tag{41}$$

Let $\epsilon_\tau^\dagger = r(s_\tau, a_\tau) + \gamma V^\dagger(s_{\tau+1}) - \mathbb{B}V^\dagger(s,a)$, we have

$$(\text{iii})^2 = \|\sum_{\tau=1}^N \phi_\tau \epsilon_\tau\|_{\Lambda^{-1}}^2 \leq 2\|\sum_{\tau=1}^N \phi_\tau \epsilon_\tau^\dagger\|_{\Lambda^{-1}}^2 + 2\|\sum_{\tau=1}^N \phi_\tau(\epsilon_\tau^\dagger - \epsilon_\tau)\|_{\Lambda^{-1}}^2$$

$$\leq 2\|\sum_{\tau=1}^N \phi_\tau \epsilon_\tau^\dagger\|_{\Lambda^{-1}}^2 + 8\gamma^2\epsilon^2 \sum_{\tau=1}^N |\phi_\tau \Lambda^{-1} \phi_\tau|$$

$$\leq 2\|\sum_{\tau=1}^N \phi_\tau \epsilon_\tau^\dagger\|_{\Lambda^{-1}}^2 + 8\gamma^2\epsilon^2 N^2/\lambda$$

It remains to bound $\|\sum_{\tau=1}^N \phi_\tau \epsilon_\tau^\dagger\|_{\Lambda^{-1}}^2$. From the assumption for data collection process, it is easy to show that $\mathbb{E}_\mathcal{D}[\epsilon_\tau \,|\, \mathcal{F}_{\tau-1}] = 0$, where $F_{\tau-1} = \sigma(\{(s_i, a_i)_{i=1}^\tau \cup (r_i, s_{i+1})_{i=1}^\tau\})$ is the $\sigma$-algebra generated by the variables from the first $\tau$ step. Moreover, since $\epsilon_\tau \leq 2V_{\max}$, we have $\epsilon_\tau$ are $2V_{\max}$-sub-Gaussian conditioning on $F_{\tau-1}$. Then we invoke Lemma C.7 with $M_0 = \lambda \cdot I$ and $M_k = \lambda \cdot I + \sum_{\tau=1}^k \phi(s_\tau, a_\tau) \phi(s_\tau, a_\tau)^\top$. For the fixed function $V : \mathcal{S} \to [0, V_{\max}]$, we have

$$\mathbb{P}_\mathcal{D}\left(\left\|\sum_{\tau=1}^N \phi(s_\tau, a_\tau) \cdot \epsilon_\tau(V)\right\|_{\Lambda^{-1}}^2 > 8V_{\max}^2 \cdot \log\left(\frac{\det(\Lambda)^{1/2}}{\delta \cdot \det(\lambda \cdot I)^{1/2}}\right)\right) \leq \delta \tag{42}$$

for all $\delta \in (0,1)$. Note that $\|\phi(s,a)\| \leq 1$ for all $(s,a) \in \mathcal{S} \times \mathcal{A}$ by Definition 3.1. We have

$$\|\Lambda\|_{\mathrm{op}} = \left\|\lambda \cdot I + \sum_{\tau=1}^N \phi(s_\tau, a_\tau)\phi(s_\tau, a_\tau)^\top\right\|_{\mathrm{op}} \leq \lambda + \sum_{\tau=1}^N \|\phi(s_\tau, a_\tau)\phi(s_\tau, a_\tau)^\top\|_{\mathrm{op}} \leq \lambda + N,$$

where $\|\cdot\|_{\mathrm{op}}$ denotes the matrix operator norm. Hence, it holds that $\det(\Lambda) \leq (\lambda + N)^d$ and $\det(\lambda \cdot I) = \lambda^d$, which implies

$$\mathbb{P}_\mathcal{D}\left(\left\|\sum_{\tau=1}^N \phi(s_\tau, a_\tau) \cdot \epsilon_\tau(V)\right\|_{\Lambda_{-1}}^2 > 4V_{\max}^2 \cdot \left(2 \cdot \log(1/\delta) + d \cdot \log(1 + N/\lambda)\right)\right)$$

$$\leq \mathbb{P}_\mathcal{D}\left(\left\|\sum_{\tau=1}^N \phi(s_\tau, a_\tau) \cdot \epsilon_\tau(V)\right\|_{\Lambda_{-1}}^2 > 8V_{\max}^2 \cdot \log\left(\frac{\det(\Lambda)^{1/2}}{\delta \cdot \det(\lambda \cdot I)^{1/2}}\right)\right) \leq \delta.$$

Therefore, we conclude the proof of Lemma C.3.

Applying Lemma C.3 and the union bound, we have

$$\mathbb{P}_\mathcal{D}\left(\sup_{V \in \mathcal{N}(\varepsilon)} \left\|\sum_{\tau=1}^N \phi(s_\tau, a_\tau) \cdot \epsilon_\tau(V)\right\|_{\Lambda^{-1}}^2 > 4V_{\max}^2 \cdot \left(2 \cdot \log(1/\delta) + d \cdot \log(1 + N/\lambda)\right)\right) \leq \delta \cdot |\mathcal{N}(\varepsilon)|. \tag{43}$$

Recall that

$$\widehat{V} \in \mathcal{V}(R_0, B_0, \lambda), \qquad \text{where } R_0 = V_{\max}\sqrt{Nd/\lambda}, \; B_0 = 2\beta, \; \lambda = 1, \; \beta = c \cdot dV_{\max}\sqrt{\zeta}. \tag{44}$$

Here $c > 0$ is an absolute constant, $\xi \in (0,1)$ is the confidence parameter, and $\zeta = \log(2dV_{\max}/\xi)$ is specified in Algorithm 2. Applying Lemma C.6 with $\varepsilon = dV_{\max}/N$, we have

$$\log|\mathcal{N}(\varepsilon)| \leq d \cdot \log(1 + 4d^{-1/2}N^{3/2}) + d^2 \cdot \log(1 + 32c^2 \cdot d^{1/2}N^2\zeta)$$

$$\leq d \cdot \log(1 + 4d^{1/2}N^2) + d^2 \cdot \log(1 + 32c^2 \cdot d^{1/2}N^2\zeta). \tag{45}$$

By setting $\delta = \xi/|\mathcal{N}(\varepsilon)|$, we have that with probability at least $1 - \xi$,

$$
\Big\| \sum_{\tau=1}^{N} \phi(s_\tau, a_\tau) \cdot \epsilon_\tau(\widehat{V}) \Big\|_{\Lambda^{-1}}^{2}
$$

$$
\leq 8 V_{\max}^2 \cdot \left(2 \cdot \log(V_{\max}/\xi) + 4d^2 \cdot \log(64c^2 \cdot d^{1/2} N^2 \zeta) + d \cdot \log(1 + N) + 4d^2\right)
$$

$$
\leq 8 V_{\max}^2 d^2 \zeta (4 + \log(64c^2)). \tag{46}
$$

Here the last inequality follows from simple algebraic inequalities. We set $c \geq 1$ to be sufficiently large, which ensures that $36 + 8 \cdot \log(64c^2) \leq c^2/4$ on the right-hand side of Equation (46). By Equations (37) and (46), it holds that

$$
|(\mathrm{ii})| \leq c/2 \cdot d V_{\max} \sqrt{\zeta} \cdot \sqrt{\phi(x,a)^\top \Lambda^{-1} \phi(s,a)} = \beta/2 \cdot \sqrt{\phi(x,a)^\top \Lambda^{-1} \phi(s,a)} \tag{47}
$$

By Equations (20), (35), (36), and (47), for all $(s,a) \in \mathcal{S} \times \mathcal{A}$, it holds that

$$
\left| (\mathbb{B}\widehat{V})(s,a) - (\widehat{\mathbb{B}}\widehat{V})(s,a) \right| \leq (V_{\max}\sqrt{d} + \beta/2) \cdot \sqrt{\phi(s,a)^\top \Lambda^{-1} \phi(s,a)} \leq \Gamma(s,a) \tag{48}
$$

with probability at least $1 - \xi$. Therefore, we conclude the proof of Lemma C.3. $\qquad\square$

**Lemma C.4** (Bounded weight of value function). *Let* $V_{\max} = r_{\max}/(1 - \gamma)$. *For any function* $V : \mathcal{S} \to [0, V_{\max}]$, *we have*

$$
\|w\| \leq V_{\max}\sqrt{d}, \|\widehat{w}\| \leq V_{\max}\sqrt{\frac{Nd}{\lambda}}.
$$

*Proof.* Since

$$
w^\top \phi(s,a) = \langle M, \phi(s,a) \rangle + \gamma \int V(s')\psi(s')^\top M \phi(s,a)\mathrm{d}s',
$$

we have

$$
\begin{aligned}
w &= M + \gamma \int V(s')\psi(s')^\top M \mathrm{d}s' \\
&= r_{\max}\sqrt{d} + \gamma V_{\max}\sqrt{d} \\
&= V_{\max}\sqrt{d}.
\end{aligned}
$$

For $\widehat{w}$, we have

$$
\begin{aligned}
\|\widehat{w}\| &= \|\Lambda^{-1} \sum_{\tau=1}^{N} \phi_\tau(r_\tau + \gamma V(s_{\tau+1}))\| \\
&\leq \sum_{\tau=1}^{N} \|\Lambda^{-1}\phi_\tau(r_\tau + \gamma V(s_{\tau+1}))\| \\
&\leq V_{\max} \sum_{\tau=1}^{N} \|\Lambda^{-1}\phi_\tau\| \\
&\leq V_{\max} \sum_{\tau=1}^{N} \sqrt{\phi_\tau^\top \Lambda^{-1/2} \Lambda^{-1} \Lambda^{-1/2} \phi_\tau} \\
&\leq \frac{V_{\max}}{\sqrt{\lambda}} \sum_{\tau=1}^{N} \sqrt{\phi_\tau^\top \Lambda^{-1} \phi_\tau} \\
&\leq V_{\max} \sqrt{\frac{N}{\lambda}} \sqrt{\mathrm{Tr}(\Lambda^{-1} \sum_{\tau=1}^{T} \phi_\tau \phi_\tau^\top)} \\
&\leq V_{\max} \sqrt{\frac{Nd}{\lambda}}.
\end{aligned}
$$

$\qquad\square$

**Lemma C.5.** *For policy $\pi^*$ and any reward function parameter $\theta \in \mathcal{C}(\delta)$, we have*

$$|V_\theta^{\pi^*} - V_{\widehat{\theta}}^{\pi^*}| \leq \frac{2r_{max}}{1-\gamma}\sqrt{\frac{d^2\zeta_2}{N_0 C_0^\dagger}}.$$

*Proof.* From the definition, we have

$$
\begin{aligned}
|r_\theta(s,a) - r_{\widehat{\theta}}(s,a)| &= |\phi(s,a)^\top\theta - \phi(s,a)^\top\widehat{\theta}| \\
&\leq \|\theta - \widehat{\theta}\|_\Lambda \cdot \|\phi(s,a)\|_{\Lambda^{-1}} \\
&\leq \alpha\sqrt{\phi(s,a)^\top\Lambda^{-1}\phi(s,a)},
\end{aligned}
$$

Where the first inequality follows from the Cauchy-Schwartz inequality and the second inequality uses the fact that $\theta \in \mathcal{C}(\delta)$. Then we have

$$
\begin{aligned}
V_\theta^{\pi^*}(s) - V_{\widehat{\theta}}^{\pi^*}(s) &= \mathbb{E}_{\pi^*}\left[\sum_{t=0}^\infty \gamma^t(r_\theta(s,a) - r_{\widehat{\theta}}(s,a)) \,\Big|\, s_0 = s\right] \\
&\leq \mathbb{E}_{\pi^*}\left[\sum_{t=0}^\infty \gamma^t|r_\theta(s,a) - r_{\widehat{\theta}}(s,a)| \,\Big|\, s_0 = s\right] \\
&\leq \alpha\mathbb{E}_{\pi^*}\left[\sum_{t=0}^\infty \gamma^t\sqrt{\phi(s,a)^\top\Lambda^{-1}\phi(s,a)} \,\Big|\, s_0 = s\right] \\
&\leq \frac{\alpha}{1-\gamma}\sqrt{\sum_{j=1}^d \frac{1}{1 + C_0^\dagger \cdot N_0}} \leq \frac{2r_{\max}}{1-\gamma}\sqrt{\frac{d^2\zeta_2}{N_0 C_0^\dagger}},
\end{aligned}
$$

where the second to last inequality follows similarly as Equation (26) (27) and the last inequality follows from the fact that $\alpha = 2r_{\max}\sqrt{d\zeta_2}$.

Note that such a choice of $\alpha$ is sufficient for Lemma 4.1 to hold since

$$
\begin{aligned}
\sqrt{\nu} + r_{\max} \cdot \sqrt{2\log\frac{1}{\delta} + d\log\left(1 + \frac{N_0}{\nu d}\right)} &= 1 + r_{\max} \cdot \sqrt{2\log\frac{1}{\delta} + d\log\left(1 + \frac{N_0}{d}\right)} \\
&\leq 1 + r_{\max} \cdot \sqrt{2\log\frac{1}{\delta} + d\log 2N_0} \\
&\leq 1 + r_{\max} \cdot \sqrt{d\log\frac{2N_0}{\delta}} \\
&\leq 2r_{\max} \cdot \sqrt{d\log\frac{2N_0}{\delta}} \\
&= 2r_{\max}\sqrt{d\zeta_2},
\end{aligned}
$$

where the inequalities holds for sufficiently small $\delta > 0$ and $d \geq 2$. $\qquad\square$

## C.1 TECHNICAL LEMMAS

**Lemma C.6** ($\varepsilon$-Covering Number (Jin et al., 2020)). *For all $h \in [H]$ and all $\varepsilon > 0$, we have*

$$\log|\mathcal{N}(\varepsilon; R, B, \lambda)| \leq d \cdot \log(1 + 4R/\varepsilon) + d^2 \cdot \log\left(1 + 8d^{1/2}B^2/(\varepsilon^2\lambda)\right).$$

*Proof of Lemma C.6.* See Lemma D.6 in Jin et al. (2020) for detailed proof. $\qquad\square$

**Lemma C.7** (Concentration of Self-Normalized Processes (Abbasi-Yadkori et al., 2011)). *Let $\{\mathcal{F}_t\}_{t=0}^\infty$ be a filtration and $\{\epsilon_t\}_{t=1}^\infty$ be an $\mathbb{R}$-valued stochastic process such that $\epsilon_t$ is $\mathcal{F}_t$-measurable*

*for all $t \geq 1$. Moreover, suppose that conditioning on $\mathcal{F}_{t-1}$, $\epsilon_t$ is a zero-mean and $\sigma$-sub-Gaussian random variable for all $t \geq 1$, that is,*

$$\mathbb{E}[\epsilon_t \,|\, \mathcal{F}_{t-1}] = 0, \qquad \mathbb{E}\big[\exp(\lambda\epsilon_t) \,\big|\, \mathcal{F}_{t-1}\big] \leq \exp(\lambda^2\sigma^2/2), \qquad \forall\lambda \in \mathbb{R}.$$

*Meanwhile, let $\{\phi_t\}_{t=1}^{\infty}$ be an $\mathbb{R}^d$-valued stochastic process such that $\phi_t$ is $\mathcal{F}_{t-1}$-measurable for all $t \geq 1$. Also, let $M_0 \in \mathbb{R}^{d \times d}$ be a deterministic positive-definite matrix and*

$$M_t = M_0 + \sum_{s=1}^{t} \phi_s \phi_s^{\top}$$

*for all $t \geq 1$. For all $\delta > 0$, it holds that*

$$\Big\| \sum_{s=1}^{t} \phi_s \epsilon_s \Big\|_{M_t^{-1}}^2 \leq 2\sigma^2 \cdot \log\Big( \frac{\det(M_t)^{1/2} \cdot \det(M_0)^{-1/2}}{\delta} \Big)$$

*for all $t \geq 1$ with probability at least $1 - \delta$.*

*Proof.* See Theorem 1 of Abbasi-Yadkori et al. (2011) for a detailed proof. $\qquad\square$

# D  EXPERIMENTAL SETTINGS

## D.1  MULTI-TASK ANTMAZE

We first divide the source dataset (e.g., antmaze-medium-diverse-v2) into the `directed` dataset or the `undirected` dataset. The trajectories in the `undirected` dataset are randomly and uniformly divided into different tasks. In contrast, the `directed` dataset is associated with the goal closest to the final state of the trajectory. Then, for each subtask (e.g., goal=(0.0, 16.0)), we relabel the corresponding `undirected` or `undirected` dataset according to the goal. Finally, we keep the rewards in the target task dataset (labeled dataset) while setting the rewards in the other task dataset to 0 (unlabeled dataset). We visualize the `directed` or `undirected` dataset in Figure 3. We can find that the distribution shift issue in the `directed` dataset is more severe than the `undirected` dataset, which further challenges the unlabeled data sharing algorithms.

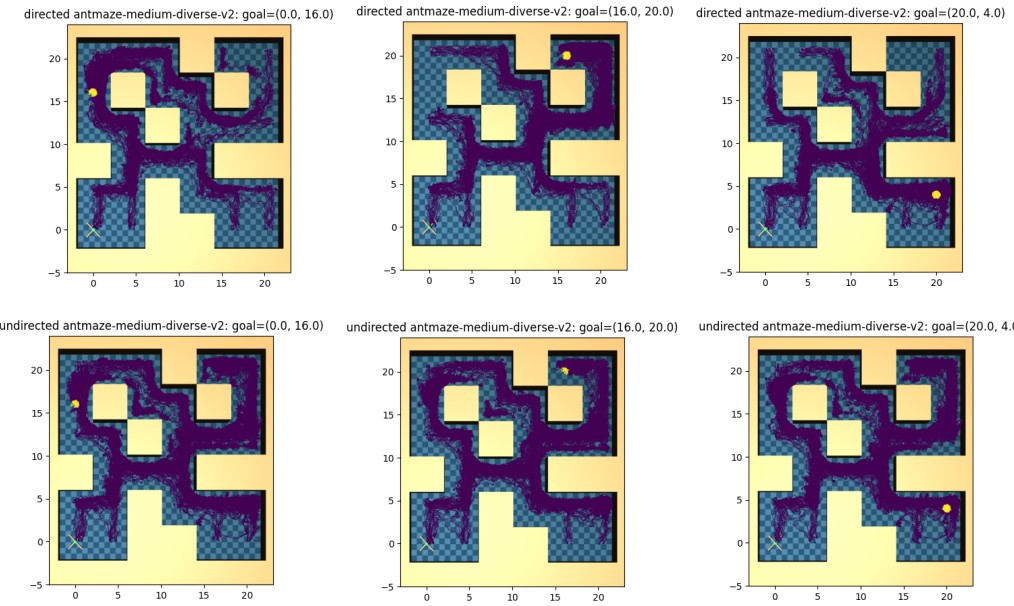

Figure 3: Visualization of multi-task antmaze-medium-diverse-v2 datasets. The purple dots denote the transition of reward 0. The yellow dots denote the transition near the goal of the sub-task and the reward is +1.

## D.2  MULTI-TASK META-WORLD

We consider the same setup as in CDS (Yu et al., 2021a) and four tasks: `door open`, `door close`, `drawer open` and `drawer close`, which is shown in Figure 4. We use medium-replay datasets with 152K transitions for task door open and drawer close and use 10 expert trajectories for task door close and drawer open.

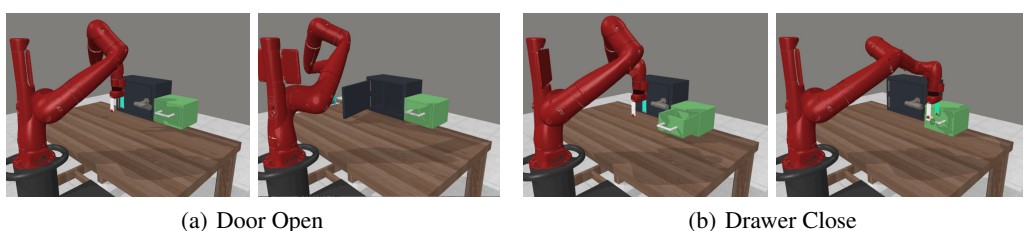

(a) Door Open          (b) Drawer Close

Figure 4: Visualization of subtasks in Meta-World.

## E    MORE DISCUSSION ON THE RELATIVE PERFORMANCE OF UDS

Based on the discussion in the main text, we can summarize the factors that affect the relative performance of PDS algorithms as follows.

| scenarios | large number of unlabeled data | high quality of unlabeled data | long horizon | large dimension |
|---|---|---|---|---|
| **finite-sample term** | ✓ | ✓ | | |
| **asymptotic term** | | | ✓ | ✓ |

Table 4: Scenarios where PDS has better relative performance.

However, it is worth noting that the asymptotic term is dependent on the backbone offline algorithm we choose. For model-based algorithms like Uehara & Sun (2021), the performance bound scale as $\mathcal{O}(d)$ and thus the problem's dimension does not affect the asymptotic term. The dependence over the discount factor is also dependent on the choice of the backbone algorithm. However, it is known that the lower bound of offline RL algorithms in linear MDPs scales as $\mathcal{O}((1-\gamma)^{1.5})$ so this asymptotic term must scales at least as $\mathcal{O}((1-\gamma)^{0.5})$ and thus negatively depends on the discount factor. That is, the larger the discount factor, the better the relative performance of PDS.

## F    DISCUSSION ON THE PERFORMANCE BOUND

### F.1    CONSTRUCTION OF ADVERSARIAL EXAMPLES

In this section, we show that there is an MDP and an "unfortunate" dataset such that the suboptimality of Algorithm 3 matches the performance bound in Theorem 4.3. We first focus on the case without data sharing. The same techniques can be used to match the reward-learning bound. We only need to show that all inequalities in Theorem 4.3 can become equality. Suppose we have an MDP with one state and $N > d$ actions. $d$ of them are optimal actions, with feature map

$$\phi(s, a_i) = (\underbrace{0, \dots}_{i-1 \text{ zeros}}, \sqrt{d}, 0, \dots)$$

and let the optimal policy be uniform over $d$ actions. Such construction makes $\Sigma_{\pi*,s} = I$ so that inequalities in Equation (25)~(27) become equalities. Then we let samples from other actions match the confidence upper bound while samples from the optimal action match the confidence lower bound, and we also need all the samples to be symmetric over different dimensions (this is required by the Cauchy-Schwitz inequality used in the proof), such that the confidence bound inequalities in Lemma C.3 also become equalities. Then, in this case, the suboptimality bound is matched and the bound in Theorem 4.3 is tight.

### F.2    REWARD BIAS OF UDS

In this section, we show that UDS suffers from a constant reward bias.

**Lemma F.1.** *By labeling all rewards in the unlabeled dataset to zeros, we have*

$$\mathbb{E}_{\mathcal{D}_0+\mathcal{D}_1}[|r(s,a) - \widehat{r}_{UDS}(s,a)|] = \frac{N_1}{N_0 + N_1} \cdot \mathbb{E}_{\mathcal{D}_1}[|r(s,a)|].$$

*That is, the reward bias does not vanish as long as the ratio of labeled data size and unlabeled data size keeps constant.*

*Proof.*

$$\mathbb{E}_{\mathcal{D}_0 + \mathcal{D}_1}[|r(s,a) - \widehat{r}_{UDS}(s,a)|] = \frac{N_0}{N_0 + N_1} \cdot \mathbb{E}_{\mathcal{D}_0}[|r(s,a) - r(s,a)|] + \frac{N_1}{N_0 + N_1} \cdot \mathbb{E}_{\mathcal{D}_1}[|r(s,a) - 0|]$$

$$= \frac{N_1}{N_0 + N_1} \cdot \mathbb{E}_{\mathcal{D}_1}[|r(s,a)|].$$

$\square$

## G    HOW BAD IS THE REWARD PREDICTION BASELINE ACTUALLY?

We conduct ablation studies for the reward prediction method from three aspects, including model size, ensemble number, and early stopping, which are shown in Table 5, Table 6 and Table 7. For the model size, 256*3 denotes the 256 hidden neurons with three hidden layers. As for the early stopping, Epoch Number = 3 denotes traversing the entire training dataset 3 times. (We find the Epoch Number=3 is enough to reduce the prediction error to a small range, and increasing epochs leads to overfitting.)

We conduct experiments in a setting where the quality of the reward labeled and reward-free data differs significantly. For example, walker2d-expert(50K)-random(0.1M) denotes 50K reward-labeled data from expert datasets and 0.1M reward-free data from random datasets. We set the PDS as the default parameter in all comparisons, including the model size 256*2, the training epoch 3, and the ensemble number 10. All experimental results adopt the normalized score metric averaged over five seeds with standard deviation.

The experimental results show that fine-tuning reward prediction can improve its performance in some cases, Nevertheless, a well-tuned reward prediction method still performs poorly compared to PDS in general. We hypothesize that this is because the "test" (reward-free) dataset may have a large distributional shift from the "training" (reward-labeled) dataset, violating the i.i.d. assumption in supervised learning.

| Model Size (Reward Prediction) | 256*3 | 512*3 | 256*4 | 512*4 | PDS |
|---|---|---|---|---|---|
| walker2d-expert(50K)-random(0.1M) | 1.8±0.3 | 11.2±2.0 | 1.2±0.2 | 13.3±2.7 | **77.6±8.1** |
| walker2d-random(50K)-expert(0.1M) | 95.1±1.9 | 96.5±1.6 | 99.4±2.0 | 97.2±2.2 | **105.1±1.2** |
| hopper-expert(50K)-random(0.1M) | 27.8±14.7 | 31.6±15.2 | 36.6±14.8 | 36.3±11.8 | **61.5±6.2** |
| hopper-random(50K)-expert(0.1M) | 72.9±19.7 | 78.9±16.7 | **92.1±15.7** | **92.3±14.6** | 93.8±4.9 |

Table 5: Ablation for the model size. We adopt various model sizes for the prediction network in the Reward Prediction baseline, while PDS adopts the default parameter model size 256*3.

| Epoch Number (Reward Prediction) | 1 | 2 | 3 | PDS |
|---|---|---|---|---|
| walker2d-expert(50K)-random(0.1M) | 2.9±1.3 | 39.6±4.1 | 1.8±0.7 | **77.6±8.1** |
| walker2d-random(50K)-expert(0.1M) | 91.2±1.3 | 101.2±1.9 | 95.1±1.6 | **105.1±1.2** |
| hopper-expert(50K)-random(0.1M) | 35.9±9.8 | 31.0±5.2 | 27.8±14.7 | **61.5±6.2** |
| hopper-random(50K)-expert(0.1M) | 42.4±11.8 | 57.8±12.2 | 84.8±10.5 | **93.8±4.9** |

Table 6: Ablation for the early stopping. We adopt various Training Epochs for the prediction network in the Reward Prediction baseline, while PDS adopts the default parameter epoch number 3.

| Environment | Reward Prediction (Ensemble=10) | PDS |
|---|---|---|
| walker2d-expert(50K)-random(0.1M) | 18.9±3.6 | **77.6±8.1** |
| walker2d-random(50K)-expert(0.1M) | 91.41±2.5 | **105.1±2.1** |
| hopper-expert(50K)-random(0.1M) | 39.4±5.7 | **61.5±6.2** |
| hopper-random(50K)-expert(0.1M) | 78.4±4.6 | **93.8±4.9** |

Table 7: Ablation for the ensemble. PDS adopts the default parameter ensemble number 10 and we adopt the same ensemble number for the prediction network in the Reward Prediction baseline.

