# OpenReview forum: "The Provable Benefit of Unsupervised Data Sharing for Offline Reinforcement Learning"
_ICLR.cc/2023/Conference — ICLR 2023 poster_

### Official Review · Reviewer_dc2J · 2022-10-22

**Confidence:** 4
**Correctness:** 3
**Technical Novelty And Significance:** 3
**Empirical Novelty And Significance:** 3
**Recommendation:** 8

**Clarity, Quality, Novelty And Reproducibility:**

The presentation is clear, high quality, and novel up to the issues raised above.

In terms of reproducibility, the code is included and hyperparameters are included in the appendix to reproduce the results, but I have not tried to run it myself.

**Strength And Weaknesses:**

### Strengths

1. The paper provides a novel and simple approach to the important problem of leveraging reward-free data for offline RL. In particular, it is novel and useful to show that a simple reward penalty can suffice to incorporate reward-free data into offline RL.

2. The paper provides a theoretical analysis of the algorithm in linear MDPs that shows the provable benefits in a simplified setting.

3. The paper also is able to scale the algorithm up to a deep RL variant that shows strong empirical performance on a substantial variety of tasks.

### Weaknesses

1. The theory is not quite complete. The proof of the main novel contribution bounding the gap between the estimated reward and true reward that yields the asymptotic term in the result is omitted "for simplicity" in Appendix B/C. I think that this shouldn't be a serious problem, as intuitively the result seems to hold, but a complete proof ought to be provided. In particular, my understanding is that the bound that is omitted is to bound the difference between the V^{\pi^*}_{\theta^*} - V^{\pi^*}_{\tilde \theta} (also there is a typo in the appendix, in the last line of the align in appendix B, it should be $ d^2$ in the first term instead of $d$ to match the main text, but this is also the term where the proof is missing). Intuitively, this can be bounded by unrolling $\pi^*$ and bounding the difference between true and estimates rewards at each step, yielding some term that depends on $ \alpha $ times the sum of the feature norms, but this needs to be worked out explicitly. It is also a bit confusing to me why so much space in the appendix is spent proving Lemmas C.1 and C.3, which seem to not be novel to this paper and could perhaps be cited from prior work(let me know if I missed some detail that makes them different).

2. There is an issue with hyperparameter tuning in the experiments. Essentially, the two main baselines are UDS and prediction which correspond to special cases of the proposed PDS algorithm (UDS is PDS with $ k = \infty$ and prediction is PDS with $k = 0$). This itself is not a problem, as it is nice to have more general algorithms with improved capabilities. The problem is that it seems that the experiments rely on (potentially extensive) tuning of the $ k$ hyperparameter for PDS while the baselines have no such hyperparameter since they correspond to special cases. First, I would like to see a better description of how the hyperparameter was tuned. Appendix G lists the final values that were used, and they seem to be unique for almost every task, suggesting that some tuning method was used, but no explanation is given. Second, I think there ought to be a more straightforward discussion in the presentation of the algorithm that PDS is a generalization/unification of UDS and prediction and that thus it is not surprising that tuning $k$ yields better results than either baseline.

**Summary Of The Paper:**

This paper proposes provable data sharing (PDS) for leveraging reward-free data for offline RL. The algorithm amounts to constructing a pessimistic estimate of the reward function, relabeling all available data with those pessimistic rewards, and then running standard offline RL. This is shown to be provably efficient in linear MDPs and extended to the deep RL setting via an ensemble of reward models to estimate the pessimistic reward. Empirical results show improvements over MLE reward models and the UDS algorithm which labels all new data with zero reward.

**Summary Of The Review:**

Overall, I really liked the core idea of the paper and think that the presentation is solid. There are two key weaknesses that I raised in my review. The issue with the theory should be easy to resolve. The issue with hyperparameter tuning seems to be somewhat more fundamental and should be addressed, but is not a fatal flaw. I will rate the paper as a weak reject for now, but am happy to improve the score to accept if the authors address the issues I raised.

---

> ### Author Response · Authors · 2022-11-14
> **Response to Reviewer dc2J (Part I)**
>
> Dear Reviewer,
>
> Thanks for finding our paper high-quality and solid.
> The main issues regarding theoretical analysis and selecting the $k$-parameter are addressed as follows.
> As part of our response, we have added supplementary experimental results on PDS with an automatic $ k$ adjustment mechanism.
>
> **W1-1: There is a typo in Appendix B, and the proof of the main theorem is incomplete.**
>
> **A for W1-1** Yes, it should be $d^2$ in the last line of the equation in Appendix B.
> Thank the Reviewer for pointing it out and providing detailed instructions.
> We have updated a complete proof with more details in Appendix B/C in the revised version.
>
> **W1-2: Why so much space in the appendix is spent proving Lemmas C.1 $\sim$ C.3, which seems not to be novel to this paper?**
>
> **A for W1-2:** The main difference is that Lemma C.1 and Lemma C.2 apply to **all** reward functions in the range $[0, r_\text{max}]$ with parameterization $r(s,a)=\langle \phi(s,a), \theta\rangle, \|\theta\|_2^2\leq d$, instead of a single given reward function as in previous papers like [1]. Although the proof structures of these lemmas are similar to prior works, since only the range of the reward function matters, they require careful treatment. Lemma C.1 and Lemma C.2 allow us to decompose the suboptimality with respect to different reward functions (as the term $\widehat{V}\_{\theta\^\star}-V\^{\widehat{\pi}}\_{\theta\^\star}$ and $V\^{\pi\^*}\_{\tilde{\theta}}-\widehat{V}\_{\tilde{\theta}}$ in Appendix B), so they are essential and can not be omitted.
>
> **W2-1: Different $k$ is used in different games. There should be a better description of how the hyperparameter was tuned.**
>
> **A for W2-1:**
> We agree with the reviewer's suggestion that there should be a better description of how the hyperparameter was tuned.
> Therefore, we first give detailed guidance on selecting $k$ as follows.
> Further, we propose a simple yet efficient automatic $k$-parameter-adjustment mechanism.
>
> **Guidance for selecting $k$-parameter:**
>
> - When the quality of the labeled and unlabeled data is similar (e.g., both labeled and unlabeled data come from the same datasets), the amount of pessimism needed is low, and we select a smaller $k$ (e.g., $k=5$).
>
>
> - When the quality of the labeled and unlabeled data differs significantly,  we select $k$ according to the quality of the unlabeled data. Specifically,
>     - When the labeled data is of higher quality than the unlabeled data (e.g., labeled data comes from Expert datasets while the unlabeled data comes from Random datasets).
>     The amount of pessimism needed is high, and we select a larger $k$ (e.g., $k=15$).
>
>     - On the contrary, when the labeled data is of lower quality than the unlabeled data (e.g., labeled data comes from Random datasets while the unlabeled data comes from Expert datasets). We select a lower $k$ (e.g., $k=10$) since we prefer the optimistic reward in the high-quality unlabeled data.

---

> > ### Author Response · Authors · 2022-11-14
> > **Response to Reviewer dc2J (Part II)**
> >
> > **Automatic $k$-parameter-adjustment mechanism:**
> >
> > We observe that the amount of pessimism needed for different domains is propositional to the difference in mean rewards between labeled and unlabeled data.
> > Based on this observation, we propose a simple yet efficient automatic $k$-parameter-adjustment mechanism as follows:
> >
> > $$
> >     \widehat{r}(s,a) = \max \\{
> >     \min\_{j=1,\ldots,L} f\_{\theta\_j}(s,a) - k \sigma(s,a),0\\},
> > $$
> >
> > $$
> > \text{where} \quad
> > k = a\cdot \frac{\max({\mu - \hat{\mu}},0)}{{|\mu|+\epsilon}},
> > $$
> >
> > where $\mu=\frac{1}{N0}\sum\_{i=1}^{N0}\mu(s\_i, a\_i),\hat{\mu}=\frac{1}{N1}\sum\_{i=1}^{N1}\hat{\mu}(s\_i, a\_i)$ are the mean reward of labeled and (predicted) unlabeled data, respectively.
> >
> > We use $a=25$ and $L=10$ in all experiments.
> > The experimental results in Table 1 demonstrate that the PDS++ (PDS with the automatic parameter-adjustment mechanism) achieves strong performance across various domains.
> > **The experimental results show that our methods with universal parameters $a$ and $L$ for adaptively adjusting $k$ significantly outperform the baselines.**
> > We have added a more detailed description in Appendix G in the revised paper.
> >
> >
> > |Tasks|UDS|Prediction|PDS++|
> > |-----|-----|-----|-----|
> > |walker2d-medium/50K-medium/0.1M | 71.0$\pm$3.9 | 71.6$\pm$2.2 | **76.1$\pm$0.2** |
> > |walker2d-medium/50K-medium/0.4M | 75.1$\pm$1.8 | 70.3$\pm$3.9 | **80.1$\pm$0.3** |
> > |walker2d-medium/50K-medium/0.6M | 74.0$\pm$0.6 | **79.8$\pm$3.6** | **79.1$\pm$1.4**|
> > |walker2d-expert/50K-random/0.1M | 17.7$\pm$12.2 | 1.8$\pm$0.7 |  **39.5$\pm$10.0**|
> > |walker2d-random/50K-expert/0.1M | 0.6$\pm$0.1 | 95.1$\pm$1.6 | **101.4$\pm$3.2** |
> > |hopper-medium/50K-medium/0.1M | 59.3$\pm$2.1 | 65.6$\pm$3.0 | **73.9$\pm$8.4** |
> > |hopper-medium/50K-medium/0.4M | 57.2$\pm$1.6 | 69.2$\pm$3.8 | **77.8$\pm$7.4** |
> > |hopper-medium/50K-medium/0.6M | 56.7$\pm$1.7 | 68.6$\pm$1.8 | **75.9$\pm$2.4** |
> > |hopper-expert/50K-random/0.1M | **52.5$\pm$4.1** | 27.8$\pm$14.7 | 42.7$\pm$9.8|
> > |hopper-random/50K-expert/0.1M | 4.0$\pm$0.5 | 84.8$\pm$10.5 | **92.3$\pm$9.8** |
> > |antmaze-medium-play (3 tasks) / directed | 15.6$\pm$1.4 | 26.8$\pm$3.1 |**40.0$\pm$3.6**|
> > |antmaze-medium-play (3 tasks) / undirected | 19.4$\pm$2.0 | **28.7$\pm$3.7** | **29.2$\pm$4.1** |
> > |antmaze-medium-diverse (3 tasks) / directed | 9.1$\pm$3.2 | 20.0$\pm$4.0 | **53.2$\pm$3.6**|
> > |antmaze-medium-diverse (3 tasks) / undirected | 9.9$\pm$1.0 | **41.9$\pm$4.6** | **42.5$\pm$5.2**|
> > |meta-world / door-open | 20.3$\pm$10.8 | **29.0$\pm$13.4** | 25.5$\pm$15.5 |
> > |meta-world / door-close | 0.0$\pm$0.0 | 109.0$\pm$15.6 | **114.3$\pm$1.8** |
> > |meta-world / drawer-open | 38.0$\pm$65.8 | 90.0$\pm$37.3 | **153.8$\pm$0.4**|
> > |meta-world / drawer-close | **182.3$\pm$0.4** | **182.3$\pm$0.4** | **182.8$\pm$0.4**
> >
> > Table 1. Comparison between baselines and PDS++ (PDS with the automatic $k$-adjustment mechanism).
> > Here we use the universal parameter $a=25, L=10$ in all domains.
> > The experimental results are the normalized score metric averaged with five random seeds.
> >
> >
> >
> > **W2-2: There should be a straightforward discussion between PDS, UDS, and Prediction.**
> >
> > **A for W2-2:**
> > Thanks for the suggestion, and we have added a detailed discussion in the revised version.
> >
> > Thanks again for the Reviewer's valuable comments.
> > We sincerely hope our response has cleared the Reviewer's concern regarding the theoretical analysis and experiments.
> >
> > Best Regards,
> >
> > The Authors
> >
> > References
> >
> > [1] Jin, Ying, Zhuoran Yang, and Zhaoran Wang. "Is pessimism provably efficient for offline rl?." International Conference on Machine Learning. PMLR, 2021.

---

> > > ### Comment · Reviewer_dc2J · 2022-11-15
> > > **Looks good**
> > >
> > > The updated paper resolves the two main issues I raised before. The added proof shores up the theoretical result and the added discussion of PDS as a generalization of UDS and prediction, along with clarified experimental protocol resolves the second. As such I have increased my score to accept.

---

> ### Author Response · Authors · 2022-11-16
> **Thanks for raising the score to 8!**
>
> We would like to thank the reviewer for raising the score! We really appreciate the valuable comments and suggestions from the reviewer.

---

### Official Review · Reviewer_ZyTC · 2022-10-24

**Confidence:** 4
**Correctness:** 3
**Technical Novelty And Significance:** 3
**Empirical Novelty And Significance:** 3
**Recommendation:** 6

**Clarity, Quality, Novelty And Reproducibility:**

Clarify -- Generally the paper is easy to follow. I have left some writing suggestions above.

Quality -- I really like the simplicity of the underlying idea. I have a few issues with some of the claims (e.g., do they actually show that unsupervised data sharing is good? do they also apply to the reward prediction baseline) that would be good to clarify.

Novelty -- The method is a novel extension of prior reward prediction methods.

Reproducibility -- The appendix does not contain details/hyperparameters, but the supplemental material contains code with documentation.

**Strength And Weaknesses:**

Strengths
* The proposed method is very simple
* Empirically, the proposed method achieves great results

Weaknesses
* I'm not sure that the main theorem actually proves the claim in the title (that unsupervised data sharing is provably useful). The issue is that Theorem 4.3 only provides a _bound_ on the suboptimality. The bound decreases when we make use of unlabeled data.
However, if the bound isn't tight, then decreasing the bound doesn't necessarily mean that the unlabeled data is useful. As a toy example, the bound $\frac{2 r_\text{max}}{1 - \gamma} + N_0 + N_1$ is also a valid (and vacuous) bound on the suboptimality. This bound _decreases_ if we reduce the amount of labeled and unlabeled data, suggesting that we should throw away all the data.
* I'm unsure whether the analysis about the suboptimality ratio in Eq 13/14 is solid, for the same reason as above. If this bound isn't tight, then this ratio might be meaningless.
* The clarity of the writing could be improved (see details below)
* I'm unsure how bad the reward prediction baseline actually is, _if it is very well tuned_. For example, it might be important to use high-a capacity prediction network; it might be important to use the same ensemble as the proposed method; it might be important to use early stopping to prevent overfitting. I think that showing that _even a very tuned_ reward prediction method can fail would strengthen this paper.


Additional comments
*  Title -- I'm not sure "Provable" is being used correctly. Typically it refers to a claim that could be proven; saying "this provable theorem" is grammatically correct, if redundant. I'd recommend changing the title to something like "The Provable Benefits of..." or "Theoretical Gaurantees for ...." or "A Theoretical Justification for ..." [Though, see comment above about the correctness of these claims.]
* "obviating" -> "reducing" -- Presumably, the method still needs some annotations.
* "merit" -- Which merit? Perhaps rewrite as "Like self-supervised methods in other areas of ML, we would like self-supervised RL methods that can ..."
* "in a supervised manner" (in abstract and intro) -- I found this statement a bit strange, as it it were arguing that offline RL were equivalent to supervised learning. Perhaps behavioral cloning methods can argue this, but I don't think most offline RL methods are "supervised" in this way.
* "is highly preferred" -- Preferred to what alternative?
* "can be costly and requires huge human efforts" -- Cite.
* "CV, NLP" -- No need to define acronyms if they aren't used again
* "which is even worse than labeling the unsupervised dataset with all zeros" -- I might be misremembering the UDS paper, but this seems like a misrepresentation of this prior work. I'd recommend a more generous statement (e.g., "Prior work has shown the learning to predict the rewards can be challenging, and that the simple approach of setting the reward to zero can achieve good results.")
* "How can we leverage unlabeled data" -- Perhaps clarify that "unlabeled" means reward-free, not action-free.
* "reduced to some sort of linear bandits" -- I didn't understand this during the first pass; perhaps add more context, or wait to introduce this to the theoretical section.
* "as shown in Algorithm 3" -- Broken reference? Algorithm 3 doesn't appear in the main text.
* "The coverage coefficient represents the maximum ratio ..." -- How does this relate to the ratio of the state occupancy measures, $\rho(s, a)$?
* "How to leverage" -> "How can we leverage"
* "we have the following lemma" -- I would highly recommend adding a few sentences describing what the result will say and providing some intuition, before stating the formal result.
* "Lemma 4.1" -> "Lemma 4.1 [Abbasi-Yadkori]" -- Make clear that this is not a contribution of this paper
* Eq. 8 -- Where is $\hat{\theta}$ defined?
* "model-based approach with different amounts of pessimism" -- This "strawman" approach is described on page 4 and in the introduction. I found it a bit odd because it was unclear whether this was referring to a specific prior method. I'd probably just cut this, and direct the comparison to be with UDS/CDS.
* "summarized in Algorithm 1" -- Add a paragraph introducing the method first. Something like "We now introduce our method. We do ... We summarize our approach in Algorithm 1."
* "set as in Equatoin 11" -- Broken reference? Eq 11 hasn't appeared yet.
* "bi-level optimization problem" -- I'd recommend writing out what this problem is. As written, Eq 11 doesn't look like a bilevel optimization problem.
* Eq. 9 -- Why do we need to clip the reward $\hat{r}$ to be positive? Is this motivated by the theory?
* Lemma 4.2 -- I'd recommend writing out the result in words, before stating the math.
* Eq. 11 -- Where is $\hat{V}$ defined? (I.e., is it the solution to some other optimization problem/procedure?)
* Theorem 4.3 -- Does this make the linear MDP assumption? If so, I'd recommend mentioning it.
* Theorem 4.3 -- Does this theorem also apply to simple (non-pessimistic) reward prediction methods?
* Theorem 4.3 -- When is this bound non-vacuous? E.g., if $\sqrt{d^2 \zeta_2 / (N_0 C_0)} > 1$, then this is vacuous; so, when will this square root be much less than 1?
* "This means that ..." -- Great intuition!
* Eq 15, Eq 16 -- I was confused which method was used in the analysis, and which was used for the main experiments.
* For all experiments, I'd recommend increasing the number of random seeds to 5.
* Table 2 -- It's a bit non-standard to show separate performance numbers for different goals. I'd recommend just showing the standard evaluation numbers, so that they are comparable with the numbers reported in prior work.
* Fig -- Increase figure resolution. What is the "Learn" line?
* Fig 2 -- I was a bit confused what to learn from this plot; at face value, it seems to indicate that UDS is the best
* "statistics(exact" -- Missing space in one of the citations.
* "Training agent for first-person" -- Missing journal/identifier in this citation.
* Table 4 -- This table seems to indicate that CDS+UDS is the strongest baseline. I'd highly recommend including the strongest baselines in the main text.
* Table 4 -- What does the "Learn" method correspond to?

**Summary Of The Paper:**

This paper proposes an offline RL method for the setting where the agent is additionally provided a dataset of reward-free transitions. Different from prior work, this method imputes the rewards for those transitions using a lower confidence bound. Empirically, the proposed method outperforms a prior method that uses 0 for these reward-free transitions. The paper provides theoretical results showing that the suboptimality of the learned policy can be decreased by using these reward-free transitions.

**Summary Of The Review:**

Overall, I think the proposed method seems quite promising -- a really simple way for making use of reward-free data. I appreciate that the paper contains theoretical results, too, to supplement the empirical results. I have a few concerns/questions noted above. If these are addressed, I will vote for accepting the paper.

---

> ### Author Response · Authors · 2022-11-14
> **Response to Reviewer ZyTC (Part I)**
>
> Dear Reviewer,
>
> Thanks for finding our work novel and quite promising.
> We provide clarification to the Reviewer's comments in the following.
> We also add supplementary ablation results on the prediction baselines as part of our response.
>
> **W1\&2: The theorem only provides a bound on the suboptimality.**
>
> **A1\&2:** The main theorem is significant because the bound is tight. We can find an "adversarial" dataset such that the suboptimality matches the bound so improving the bound means improving the worst-case performance. When making the returns in the dataset match the lower confidence bound, the inequality in Lemma C.3 becomes equality, and the bound becomes tight. In the revised version, we provide more details for such example construction in Appendix D.
> The analysis of the suboptimality ratio is also significant in a similar sense, which means that there is an "adversarial" dataset (against no data-sharing algorithm) such that the performance ratio is at least as small as in Corollary 4.4.
> We have made these points more apparent in the revised version.
>
> **W4: How bad is the reward prediction baseline actually?**
>
> **A4:** According to the reviewer's suggestions, we conduct ablation studies for the reward prediction method from three aspects, including model size, early stopping, and ensemble number, which are shown in Table 1-3.
> We conduct experiments on the Mujoco tasks, and all experimental results adopt the normalized score metric averaged over five seeds with standard deviation.
>
> The experimental results show that fine-tuning reward prediction can sometimes improve performance (e.g., high capacity in the hopper-random-expert task and early stopping in the walker2d-expert-random task).
> Nevertheless, a well-tuned reward prediction method still performs poorly compared to PDS in general.
> We hypothesize that this is because of the distributional shift between the "test" (reward-free) and "training" (reward-labeled) dataset, which is not necessarily i.i.d. in the offline RL setting.
> In the revised version, we added more details for the experiment in Appendix H.
>
>
> | Model Size (Reward Prediction) | 256*3 | 512*3 | 256*4 | 512*4 | PDS |
> | :---: | :---: | :---:|  :---:|  :---:|  :---:|
> |walker2d-expert(50K)-random(0.1M) | 1.8$\pm$0.3 | 11.2$\pm$2.0 | 1.2$\pm$0.2 | 13.3$\pm$2.7 | **77.6$\pm$8.1**|
> |walker2d-random(50K)-expert(0.1M)| 95.1$\pm$1.9 | 96.5$\pm$1.6 | 99.4$\pm$2.0 | 97.2$\pm$2.2 | **105.1$\pm$1.2** |
> |hopper-expert(50K)-random(0.1M)| 27.8$\pm$14.7 | 31.6$\pm$15.2 | 36.6$\pm$14.8 | 36.3$\pm$11.8 | **61.5$\pm$6.2** |
> |hopper-random(50K)-expert(0.1M)| 72.9$\pm$19.7 | 78.9$\pm$16.7 | **92.1$\pm$15.7** | **92.3$\pm$14.6** | **93.8$\pm$4.9** |
>
> Table 1. Ablation for the model size. 256\*3 denotes the 256 hidden neurons with three hidden layers.
> We adopt various model sizes for the prediction network in the Reward Prediction baseline, while PDS adopts the default parameter model size 256*3.
>
>
>
> |Epoch Number (Reward Prediction) | 1 | 2 | 3 | PDS |
> | :---: | :---: | :---:|  :---:|  :---:|
> |walker2d-expert(50K)-random(0.1M)| 2.9$\pm$1.3 | 39.6$\pm$4.1 | 1.8$\pm$0.7 | **77.6$\pm$8.1** |
> |walker2d-random(50K)-expert(0.1M)|91.2$\pm$1.3|101.2$\pm$1.9| 95.1$\pm$1.6|**105.1$\pm$1.2**|
> |hopper-expert(50K)-random(0.1M)| 35.9$\pm$9.8 | 31.0$\pm$5.2 | 27.8$\pm$14.7 | **61.5$\pm$6.2**|
> |hopper-random(50K)-expert(0.1M)| 42.4$\pm$11.8 | 57.8$\pm$12.2 | 84.8$\pm$10.5 | **93.8$\pm$4.9**|
>
> Table 2. Ablation for the early stopping.
> Epoch Number = 3 denotes traversing the entire training dataset 3 times. (We find the Epoch Number=3 is enough to reduce the prediction error to a small range, and increasing epochs leads to overfitting.)
> We adopt various Training Epochs for the prediction network in the Reward Prediction baseline, while PDS adopts the default parameter epoch number 3.
>
>
>
> | Environment | Reward Prediction (Ensemble=10) | PDS |
> | :---: | :---: | :---:|
> |walker2d-expert(50K)-random(0.1M) | 18.9$\pm$3.6 | **77.6$\pm$8.1**|
> |walker2d-random(50K)-expert(0.1M) | 91.41$\pm$2.5 | **105.1$\pm$2.1**|
> |hopper-expert(50K)-random(0.1M) | 39.4$\pm$5.7 | **61.5$\pm$6.2**|
> |hopper-random(50K)-expert(0.1M) | 78.4$\pm$4.6 | **93.8$\pm$4.9** |
>
> Table 3. Ablation for the ensemble.
> PDS adopts the default parameter ensemble number 10, and we adopt the same ensemble number for the prediction network in the Reward Prediction baseline.

---

> > ### Author Response · Authors · 2022-11-14
> > **Response to Reviewer ZyTC (Part II)**
> >
> > **W3: The clarity of the writing and additional comments.**
> >
> > **A3:** We appreciate the reviewer for the valuable and detailed comments.
> > In the following, We summarize the changes and the responses to the detailed questions.
> >
> >
> > -------------------------------
> >
> > **C1$\sim$11,13$\sim$15,17$\sim$20,22,33,34: Writing Suggestions.**
> >
> > **A for C1$\sim$11,13$\sim$15,17$\sim$20,22,33,34:**
> > We thank the reviewer for the detailed suggestions, and we have revised the manuscript accordingly to improve the presentation from three aspects:
> >
> > - **Description:** As suggested, we have revised some words and sentences in the manuscript, which makes the description clearer.
> >
> > - **Reference:** We have added missing references and fixed broken references.
> >
> > - **Additional Explanation:** Before the lemmas and theorems, we have added more explanations. Before presenting the Algorithm, we add a paragraph introducing our method.
> >
> > **C12: The relation between Definition 3.3 and the ratio of the state occupancy measures $\rho(s, a)$?**
> >
> > **A for C12:** Definition 3.3 extends the ratio of the state occupancy measures in the linear MDP settings.
> > In tabular MDPs with the canonical one-hot representation (i.e., $\phi(s,a)=\mathbf{e}\_{(s,a)}$, where $\mathbf{e}\_{(s,a)}$ is a unit vector in  $\mathbb{R}^{|\mathcal{S}|\times|\mathcal{A}|}$, the definition is the same as the maximum ratio of the state occupancy measures.
> > As for the general linear MDPs, the state (and action) space may be infinite, so we instead consider the density of the feature vector (i.e., eigenvalues of $\mathbb{E}_\pi[\phi(s, a)\phi(s, a)^{\top}]$), which leads to Definition 3.3.
> >
> > **C16: Where is $\widehat{\theta}$ defined?**
> >
> > **A for C16:** $\widehat{\theta}$ is defined in Equation 7.
> >
> > **C21: Why do we need to clip the reward $\widehat{r}$ to be positive?**
> >
> > **A for C21:**
> > This is required in our assumption in the first paragraph of Section 3.1, where $r$ should be in the range $[0,r_\text{max}]$.
> > This assumption can be easily extended to negative reward settings.
> > Intuitively, we need the reward function to be bounded in the given range to have a bound for the variance of the value estimators.
> >
> > **C23: Where is $\widehat{V}$ defined?**
> >
> > **A for C23:** It is defined as the output of Algorithm 3 in Appendix A. We have made it clear in the updated manuscript.
> >
> > **C24$\sim$26: Questions about the Theorem 4.3.**
> >
> > **A for C24$\sim$26:**  We answer C24$\sim$26 respectively as follows:
> >
> > - **Is the Linear MDP assumption required?** Yes, linear MDP assumption is required for Theorem 4.3 to hold.
> >
> > - **Can apply to simple reward prediction methods?** Theorem 4.3 does not apply to the simple reward prediction method since it ruins the pessimistic property of the algorithm.
> > For example, let us consider a one-state MDP with $N$ actions.
> > Among these actions, only one action has a mean reward of one, while others have a mean reward of zero.
> > When $N$ is large, there may be some actions with the mean reward of zero that (occasionally) samples a high reward.
> > To solve this issue, offline RL adds uncertainty penalties to avoid such outlier samples.
> > However, using simple prediction methods for unseen actions may give a high predicted reward for such actions, and the pessimism is ruined, which leads to a loss of performance guarantees.
> > On the contrary, our method ensures that the prediction does not overestimate with high probability.
> > On the other hand, mathematically, Lemma C.3 no longer holds for the simple reward prediction method due to the missing uncertainties penalty, so the theorem does not apply.
> >
> > - **When is the bound non-vacuous?** When the dataset has a reasonable coverage coefficient $C\_0, C\_1$ and is of reasonable size, i.e., $N0 >> d^2, N_1 >> d^3$, then the bound is meaningful and not vacuous.
> > This requirement can be met in many real problems.
> > For example, in control problems like Mujoco tasks, $d$ is in the order of tens to a hundred, and 10k labeled data and 1M reward-free data should suffice for a non-vacuous bound.

---

> > > ### Author Response · Authors · 2022-11-14
> > > **Response to Reviewer ZyTC (Part III)**
> > >
> > > **C28: For Eq 15 and Eq 16, which method was used in the analysis and experiments?**
> > >
> > > **A for C28:**
> > > Both Equation 15 and Equation 16 can be considered as implementation methods of uncertainty penalty in Equation 9. Equation 15 is directly motivated by our analysis, while Equation 16 is simpler to implement. We use Equation 15 for the main experiments. Both methods achieve similar performances, as shown in Table 4 as follows.
> > >
> > > |Tasks|PDS (Eq 15) | PDS (Eq 16) |
> > > | :---: | :---: | :---:|
> > > |walker2d-medium/50K-medium/0.1M|74.0$\pm$1.3 | 76.5$\pm$2.0|
> > > |walker2d-medium/50K-medium/0.4M | 80.1$\pm$1.6 | 79.9$\pm$0.6|
> > > |walker2d-medium/50K-medium/0.6M | 81.7$\pm$0.8 | 80.1$\pm$0.6|
> > > |hopper-medium/50K-medium/0.1M | 63.6$\pm$1.6 | 75.3$\pm$1.19|
> > > |hopper-medium/50K-medium/0.4M | 68.8$\pm$1.9 | 80.7$\pm$0.92|
> > > |hopper-medium/50K-medium/0.6M | 68.7$\pm$0.5 | 79.2$\pm$3.74|
> > >
> > > Table 4. Ablation for the various pessimistic estimation of PDS.
> > >
> > >
> > >
> > > **C29$\sim$32,35,36: Questions about the experimental results.**
> > >
> > > **A for C29$\sim$32,35,36:** We answer these comments as follows:
> > >
> > > - **The number of random seeds:** As suggested, we have updated our experimental results over the 5 random seeds.
> > >
> > > - **Standard presentation on antmaze tasks:** We have revised Table 2 in the updated manuscript by showing the standard evaluation numbers and including the CDS+UDS as our baseline.
> > >
> > > - **What does the 'Learn' method correspond to?** The "Learn" method in Figure 1 and Table 4 refers to the simple reward prediction method. We have updated the manuscript to make it clearer.
> > >
> > > - **Why UDS achieves the lowest variance in Fig 2?**
> > > Fig 2 aims to show the variance of different methods.
> > > PDS achieves similar variance as the data sharing with the true reward method, which hints that PDS has a good trade-off between variance and bias.
> > > It is not surprising that UDS achieves a lower variance because it induces a high bias since UDS labels all reward-free data with zeros.
> > > We plot the reward prediction error in the following table to demonstrate this further.
> > >
> > > |Method|  UDS| PDS| Rew Pred.|
> > > | :---: | :---: | :---:|:---:|
> > > |Reward Prediction Error| 0.49$\pm$0.21 | 0.26$\pm$0.12 | 0.15$\pm$ 0.03|
> > >
> > > Table 5. Reward prediction error for different methods on metaworld tasks. The error refers to the mean absolute difference between learned and oracle rewards. The result is averaged over four environments and five seeds.
> > >
> > >
> > > Thanks again for the detailed and supportive suggestions, which significantly improved our paper.
> > > We sincerely hope that our response has cleared the Reviewer's concerns.
> > >
> > > Best Regards,
> > >
> > > The Authors

---

> > ### Comment · Reviewer_ZyTC · 2022-11-14
> > **Reviewer response (to I, II, III)**
> >
> > Dear authors,
> >
> > Thanks for the clarifications and revisions. This has addressed my concern about the reward prediction baseline. I'm still a little concerned about the tightness of bound because most MDPs won't be the worst-case MDP. I agree with the other reviewers that automatically selecting $k$ seems important, and so I appreciate this new extension by the authors. I am increasing my review 5 -> 6.

---

> > > ### Author Response · Authors · 2022-11-16
> > > **Thanks for raising the score and for further feedback!**
> > >
> > > Dear Reviewer,
> > >
> > > We thank the reviewer for raising the score!
> > > We also appreciate the valuable comments, which helped us improve the paper's strengths significantly.
> > >
> > > We would like to clarify Theorem 4.3 from the following aspects and hope that this can address the reviewer's remaining concerns.
> > >
> > > - As the reviewer pointed out, Theorem 4.3 can be used to show that additional reward-free data can improve the (worst-case) bound, but it is not our primary focus. More importantly, **this theorem indicates that our method has a performance bound that is provably better than existing methods.** This is because these methods suffer from a constant suboptimality even given sufficient data, while ours' suboptimality bound decreases as the number of data increases (the second term of Theorem 4.3). Specifically,
> > >
> > >     - A simple prediction method suffers from the distributional shift between labeled and unlabeled datasets. Such a distributional gap causes a reward prediction error proportional to the distance between the two distributions regardless of the size of the dataset [1].
> > >     - Labeling all reward-free data with zeros also suffers from a constant reward bias, since the average reward of the unlabeled dataset is not zero in general. If the ratio of labeled and unlabeled data keeps constant, the bias remains even if the total amount of data goes to infinity. We provide more details in Appendix I.
> > >
> > > Thus, other methods' suboptimality is provably larger than ours, given sufficient data. Such a conclusion holds in general and is not limited to the worst case. We have made this clearer in the revised version.
> > >
> > > - Experiments verify that we can improve over no data-sharing algorithms given additional data, as illustrated in the paper. This means that empirically, our method does benefit from reward-free data in average cases.
> > >
> > > Thanks again for the valuable comments.
> > > We sincerely hope our additional explanation has cleared the remaining concerns regarding the theoretical analysis.
> > > We are looking forward to more discussions!
> > >
> > > Best,
> > >
> > > The Authors.
> > >
> > > References
> > >
> > > [1] Ben-David, Shai, et al. "A theory of learning from different domains." Machine learning 79.1 (2010): 151-175.

---

> > > > ### Comment · Reviewer_ZyTC · 2022-11-16
> > > > **Reviewer response**
> > > >
> > > > Thanks for clarifying Theorem 4.3. A few questions to clarify my understanding:
> > > >
> > > > > Theorem 4.3  ... it is not our primary focus
> > > >
> > > > What is the primary focus? I thought this was the theorem that proved the claim in the title, but perhaps I missed another, more important result.
> > > >
> > > > > other methods' suboptimality is provably larger than ours, given sufficient data
> > > >
> > > > To clarify, the results are that the _bound_ on suboptimality is provably better, right? If $A > 5$ and $B > 10$, then we can't immediately conclude $B > A$ (except for the case where the bound is tight, as the authors brought up earlier).
> > > >
> > > > (Also, to clarify, I do think this paper should be accepted.)

---

> > > > > ### Author Response · Authors · 2022-11-18
> > > > > **Further Response to Reviewer ZyTC**
> > > > >
> > > > > Thanks for the quick reply.
> > > > > We want to clarify the remaining two questions as follows.
> > > > >
> > > > > **Q1:What is the primary focus of Theorem 4.3?**
> > > > >
> > > > > **A for Q1:**
> > > > > The primary focus of Theorem 4.3 is that our method has a performance bound that is better than existing methods, as shown in the previous reply. As a corollary, this theorem can also show that unsupervised data is provably useful (using the PDS algorithm), as the reviewer points out.
> > > > >
> > > > > **Q2: Do results prove that the *bound* on suboptimality is better?**
> > > > >
> > > > > **A for Q2:**
> > > > > Yes, we prove that the suboptimality bound is better, not the suboptimality. We agree with the reviewer that a better bound on suboptimality does not necessarily mean a better suboptimality. Unfortunately, it is generally not possible to provide theoretic proof that the suboptimality of one algorithm is better than another in such a "point-wise" manner.
> > > > > For example, suppose an algorithm uses no data and chooses a random policy. In that case, it is possible that it luckily chooses the optimal policy, so no algorithm can provably outperform this "random" algorithm. However, this algorithm can not avoid cases where the suboptimality is extremely large, so we do not use this algorithm.
> > > > > On the other hand, it is possible yet incredibly challenging to conduct an average-case analysis on suboptimality for RL algorithms, which may be an interesting future direction.
> > > > > Therefore, most existing RL literature [1,2,3,4,5,6,7,8] considers the suboptimality bound in the theoretical analysis and often yields good empirical results. For example, CQL [7] and MOPO [8] are designed based on the suboptimality bound analysis, which performs well empirically.
> > > > >
> > > > > We thank the reviewer again for the constructive discussions, and we hope our response clarifies the reviewer's concerns.
> > > > >
> > > > > References
> > > > >
> > > > > [1] Cai, Qi, et al. "Provably efficient exploration in policy optimization." International Conference on Machine Learning. PMLR, 2020.
> > > > >
> > > > > [2] Jin C, Yang Z, Wang Z, et al. "Provably efficient reinforcement learning with linear function approximation." Conference on Learning Theory. PMLR, 2020: 2137-2143.
> > > > >
> > > > > [3] Zhou, Dongruo, Jiafan He, and Quanquan Gu. "Provably efficient reinforcement learning for discounted mdps with feature mapping." International Conference on Machine Learning. PMLR, 2021.
> > > > >
> > > > > [4] Ayoub, Alex, et al. "Model-based reinforcement learning with value-targeted regression." International Conference on Machine Learning. PMLR, 2020.
> > > > >
> > > > > [5] Jin, Ying, Zhuoran Yang, and Zhaoran Wang. "Is pessimism provably efficient for offline rl?." International Conference on Machine Learning. PMLR, 2021.
> > > > >
> > > > > [6] Uehara, Masatoshi, and Wen Sun. "Pessimistic model-based offline reinforcement learning under partial coverage." arXiv preprint arXiv:2107.06226 (2021).
> > > > >
> > > > > [7] Kumar, Aviral, et al. "Conservative q-learning for offline reinforcement learning." Advances in Neural Information Processing Systems 33 (2020): 1179-1191.
> > > > >
> > > > > [8] Yu, Tianhe, et al. "Mopo: Model-based offline policy optimization." Advances in Neural Information Processing Systems 33 (2020): 14129-14142.

---

> > > > > > ### Comment · Reviewer_ZyTC · 2022-11-18
> > > > > > **Reviewer response**
> > > > > >
> > > > > > Thanks for clarifying this.

---

### Official Review · Reviewer_5Qve · 2022-10-25

**Confidence:** 3
**Correctness:** 3
**Technical Novelty And Significance:** 2
**Empirical Novelty And Significance:** 2
**Recommendation:** 6

**Clarity, Quality, Novelty And Reproducibility:**

-Some minor questions for Clarity:
1. In Definition 3.1, the episodic MDP should be finite as $(S,A,P,r, H)$, so it actually means discounted infinite-horizon MDP?
2. In Page 3: what is $\psi$ under equation 1?
3. In Equation 3: what is the $s'$ come from? Is it determined by $(s,a)$?
4. Does $\Lambda$ under Equation 9 need to be normalized by $N_0$?
5. What is $k$ in Equation 15?

-The writing is clear and easy to follow and the experimental results are provided with a lot of details and seem easy to reproducibility.

-For originality, it seems the main contribution is the design of the reward. However, it seems the performance is not convincing enough since different parameters are required in different games.

**Strength And Weaknesses:**

Strength:
1. It proposes a different reward estimation method to keep the pessimistic property of the algorithm and evaluate the proposed method PDS to 3 baselines in many different experimental settings to show the performance.
2. It has a reasonable theoretical analysis of the intuition of why data-sharing works better than no data-sharing in linear MDP cases.

Weaknesses:
1. The theoretical results do not support the main contribution of this work. It seems the main contribution of this work is the designing of the estimated reward minus a confident interval and it works better than other data-sharing methods such as letting the reward be 0 or using a learned reward function to label the reward in the extra dataset. So the theoretical results should support that the proposed reward design in this work will have better results than other data-sharing methods. However, the theoretical results only support that with reward designing, it works better than no data-sharing.
2.  Seeing the main contribution of changing $r(s,a)$ to the reward designing in Equation 15, we know that the proposed reward has an additional parameter $k$ that can be tuned with comparisons to other baselines. But from my understanding, $k$ should keep the same for all the experiments to show that there exists some universal constant that will work in a lot of settings but not need to be tuned a lot for all games. However, it seems different $k$ is used in different games when the proposed methods is conducted to compare with other baselines. This seems not fair. For instance, DQN use the same parameters for all 57 Atari games and works well [1].

[1] Mnih, Volodymyr, et al. "Human-level control through deep reinforcement learning." nature 518.7540 (2015): 529-533.


**Summary Of The Paper:**

This work target the setting of offline reinforcement learning with some extra data (without reward) sharing setting. This work proposes a revised reward function to keep the pessimistic property of the algorithm and design the algorithm in linear MDP and general MDP cases. Through different experiments, it shows the performance is better than baselines with some other data-sharing methods.

**Summary Of The Review:**

This is a very interesting work that targets data sharing in offline RL. However, the experimental performance of the proposed main technical reward design is not convincing enough to show that it can work better than other baselines since it uses different parameters $k$ in different games. The experiments didn't show that there exists a universal $k$ that works generally better than the baselines.

---

> ### Author Response · Authors · 2022-11-14
> **Response to Reviewer 5Qve (Part I)**
>
> Dear Reviewer:
>
> We thank the Reviewer for the valuable comments.
> We hope the following clarification addresses the Reviewer's concerns.
> As part of our response, we have added supplementary experimental results on PDS with an automatic $k$-adjustment mechanism.
>
> **W1: Theoretical results do not support the main contribution of this work.**
>
> **A for W1:**
> We respectfully disagree with this statement. Our method enjoys a provable performance guarantee based on our theoretical analysis.
> Such a performance bound decreases as the data increases (especially the second term in Theorem 4.3), while other methods suffer from a constant suboptimality even given sufficient data due to reward bias. Specifically,
>
> - A simple prediction method suffers from the distributional shift between labeled and unlabeled datasets. Such a distributional gap causes a reward prediction error proportional to the distance between the two distributions regardless of the size of the dataset [1].
>
> - Labeling all reward-free data with zeros also suffers from a constant reward bias, since the average reward of the unlabeled dataset is not zero in general. If the ratio of labeled and unlabeled data keeps constant, the bias remains even if the total amount of data goes to infinity. We provide more details in Appendix I.
>
>
> Thus, other methods' suboptimality is provably larger than ours, given sufficient data, which makes a stark contrast between our method and existing methods. We have made this clearer in the revised version.
>
> **W2: Different $k$ is used in different games.**
>
> **A for W2:**
> We agree with the Reviewer's suggestion that there should be a universal parameter in all domains that generally works better than the baselines. We observe that the amount of pessimism needed for different domains is propositional to the difference in mean rewards between labeled and unlabeled data.
> Based on this observation, we propose a simple yet efficient automatic $k$-parameter-adjustment mechanism as follows:
>
> $$
>     \widehat{r}(s,a) = \max \\{
>     \min\_{j=1,\ldots,L} f\_{\theta\_j}(s,a) - k \sigma(s,a),0\\},
> $$
>
> $$
> \text{where} \quad
> k = a\cdot \frac{\max({\mu - \hat{\mu}},0)}{{|\mu|+\epsilon}},
> $$
>
> where $\mu=\frac{1}{N0}\sum\_{i=1}^{N0}\mu(s\_i, a\_i)$ and $\hat{\mu}=\frac{1}{N1}\sum\_{i=1}^{N1}\hat{\mu}(s\_i, a\_i)$ are the mean rewards of labeled and (predicted) unlabeled data, respectively.
>
> We use $a=25$ and $L=10$ in all experiments.
> The experimental results in Table 1 demonstrate that the PDS++ (PDS with the automatic parameter-adjustment mechanism) achieves strong performance across various domains with universal parameters $a$ and $L$.
>
> | Tasks | UDS | Prediction | PDS++ |
> |-----|-----|------|-----|
> |walker2d-medium/50K-medium/0.1M | 71.0$\pm$3.9 | 71.6$\pm$2.2 | **76.1$\pm$0.2** |
> |walker2d-medium/50K-medium/0.4M | 75.1$\pm$1.8 | 70.3$\pm$3.9 | **80.1$\pm$0.3** |
> |walker2d-medium/50K-medium/0.6M | 74.0$\pm$0.6 | **79.8$\pm$3.6** | **79.1$\pm$1.4**|
> |walker2d-expert/50K-random/0.1M | 17.7$\pm$12.2 | 1.8$\pm$0.7 |  **39.5$\pm$10.0**|
> |walker2d-random/50K-expert/0.1M | 0.6$\pm$0.1 | 95.1$\pm$1.6 | **101.4$\pm$3.2** |
> |hopper-medium/50K-medium/0.1M | 59.3$\pm$2.1 | 65.6$\pm$3.0 | **73.9$\pm$8.4** |
> |hopper-medium/50K-medium/0.4M | 57.2$\pm$1.6 | 69.2$\pm$3.8 | **77.8$\pm$7.4** |
> |hopper-medium/50K-medium/0.6M | 56.7$\pm$1.7 | 68.6$\pm$1.8 | **75.9$\pm$2.4** |
> |hopper-expert/50K-random/0.1M | **52.5$\pm$4.1** | 27.8$\pm$14.7 | 42.7$\pm$9.8|
> |hopper-random/50K-expert/0.1M | 4.0$\pm$0.5 | 84.8$\pm$10.5 | **92.3$\pm$9.8** |
> |antmaze-medium-play (3 tasks) / directed | 15.6$\pm$1.4 | 26.8$\pm$3.1 |**40.0$\pm$3.6**|
> |antmaze-medium-play (3 tasks) / undirected | 19.4$\pm$2.0 | **28.7$\pm$3.7** | **29.2$\pm$4.1** |
> |antmaze-medium-diverse (3 tasks) / directed | 9.1$\pm$3.2 | 20.0$\pm$4.0 | **53.2$\pm$3.6**|
> |antmaze-medium-diverse (3 tasks) / undirected | 9.9$\pm$1.0 | **41.9$\pm$4.6** | **42.5$\pm$5.2**|
> |meta-world (door-open) | 20.3$\pm$10.8 | **29.0$\pm$13.4** | 25.5$\pm$15.5 |
> |meta-world (door-close) | 0.0$\pm$0.0 | 109.0$\pm$15.6 | **114.3$\pm$1.8** |
> |meta-world (drawer-open) | 38.0$\pm$65.8 | 90.0$\pm$37.3 | **153.8$\pm$0.4**|
> |meta-world (drawer-close) | **182.3$\pm$0.4** | **182.3$\pm$0.4** | **182.8$\pm$0.4**|
>
> Table 1. Comparison between baselines and PDS++ (PDS with the automatic $k$-adjustment mechanism). Here we use the universal parameter $a=25, L=10$ in all domains. The experimental results are the normalized score metric averaged with five random seeds.

---

> > ### Author Response · Authors · 2022-11-14
> > **Response to Reviewer 5Qve (Part II)**
> >
> > **W3: (Originality) This paper's main contribution is the reward's design, but different parameters are required in different games.**
> >
> > **A for W3:**
> > We respectively disagree that the main contribution is only the design of the reward for the following reasons:
> >
> > - A rigorous theoretical analysis of the benefits of reward-free data in offline RL is one of the main contributions of this work, in addition to a method to leverage such unlabelled data with theoretical guarantees.
> > The reward design is not our primary focus, and we deliberately keep our method simple without fancy uncertainty estimation methods.
> >
> > - Our analysis reveals an interesting insight that the key to properly leveraging reward-free data is to keep the pessimistic property. We also give a detailed analysis of how the factors like the discount factor, the dimension of the problem, and the number of data affect the relative performance of data-sharing algorithms compared to no data-sharing ones.
> > Such analyses are novel and interesting to the community.
> >
> > - As to different $k$ used in different domains, we show that there exist universal parameters that work for all domains after normalization, as discussed in **A for W2**.
> >
> > **Q1\&2: Do we consider the discounted infinite-horizon MDP? What is $\psi$ under Definition 3.1?**
> >
> > **A for Q1\&2:** Yes, we consider infinite-horizon discounted MDP, as mentioned in the first paragraph of Section 3.1, and $\psi$ is unnecessary in the definition.
> > Thanks for pointing out these typos; we have improved the current version to make it more transparent.
> >
> > **Q3: Does $s'$ in Equation 3 is determined by $(s,a)$?**
> >
> > **A for Q3:** Yes, The expectation is taken over the transition probability, so we have $s'\sim p(\cdot|s, a)$.
> > We have updated the equation to make it more transparent.
> >
> > **Q4: Should $\Lambda$ in Equation 9 be normalized?**
> >
> > **A for Q4:** No. $\Lambda$ is the Gram matrix of the features and is commonly used in linear bandits and linear MDP literature [2,3].
> > We should not normalize it because $\Lambda^{-1}$ reflects the uncertainty of the least-square estimator, so its eigenvalues should naturally decrease as the number of samples increases.
> >
> > **Q5: What is $k$ in Equation 15?**
> >
> > **A for Q5:** $k$ is a hyperparameter to control the amount of pessimism.
> >
> > Thanks again for the valuable comments.
> > We hope our additional experimental results and clarification address the Reviewer's concerns.
> > We sincerely hope the Reviewer can re-evaluate our paper based on our response.
> > Additional feedback is much welcomed.
> >
> > Sincerely,
> >
> > The Authors
> >
> > References
> >
> > [1] Ben-David, Shai, et al. "A theory of learning from different domains." Machine learning 79.1 (2010): 151-175.
> >
> > [2] Jin, Chi, et al. "Provably efficient reinforcement learning with linear function approximation." Conference on Learning Theory. PMLR, 2020.
> >
> > [3] Abbasi-Yadkori, Yasin, Dávid Pál, and Csaba Szepesvári. "Improved algorithms for linear stochastic bandits." Advances in neural information processing systems 24 (2011).

---

> ### Author Response · Authors · 2022-11-22
> **Looking forward to further comments!**
>
> Dear Reviewer,
>
> We have updated our supplementary experimental results about our methods with a universal parameter. We have also explained why our method is superior to other methods from the theoretical aspect. We are wondering if our response and revision have cleared your concerns. We would appreciate it if you could kindly let us know whether you have any other questions. We are looking forward to comments that can further improve our current manuscript. Thanks!
>
> Best regards,
>
> The Authors

---

> ### Author Response · Authors · 2022-12-06
> **Thanks for raising the score to 6!**
>
> We would like to thank the reviewer for raising the score! We really appreciate the valuable comments and suggestions from the reviewer.

---

### Author Response · Authors · 2022-11-14
**General Remark**

Dear Reviewers:

We first thank all the reviewers for their constructive and valuable comments. We are encouraged that the reviewers are quite positive and find our work "very interesting," "quite promising," "novel extension," "high-quality," and "solid presentation," in general.

We also noticed that reviewers were concerned about selecting the $k$-parameter, insufficient explanation of the theoretical analysis, and comparison with baselines.
We have refined the submission by adding more detailed explanations of the theoretical analysis.
We have also shown that our method with a universal parameter outperforms baselines based on a simple yet efficient $k$-parameter-adjustment mechanism.
Furthermore, we have updated the experimental results as suggested by the reviewers.

We sincerely hope that our response would address the reviewers' concerns.
Further feedback and discussions are appreciated.

Sincerely,

The Authors

---

### Decision · Program_Chairs · 2023-01-20

**Decision:**

Accept: poster

**Justification For Why Not Higher Score:**

 At least two reviewers mentioned the issues with different hyperparameters for different environments. Also two reviewers mentioned that "the theory is not quite complete." Therefore, the reviewers are not as excited by the paper as the averaging score indicates.

**Justification For Why Not Lower Score:**

Strength:

---"It proposes a different reward estimation method to keep the pessimistic property of the algorithm and evaluate the proposed method PDS to 3 baselines in many different experimental settings to show the performance."
--- It has a reasonable theoretical analysis of the intuition of why data-sharing works better than no data-sharing in linear MDP cases.



**Metareview: Summary, Strengths And Weaknesses:**

Strength:

---"It proposes a different reward estimation method to keep the pessimistic property of the algorithm and evaluate the proposed method PDS to 3 baselines in many different experimental settings to show the performance."
--- It has a reasonable theoretical analysis of the intuition of why data-sharing works better than no data-sharing in linear MDP cases.


Weakness:

---"The clarity of the writing could be improved"

--- "The theory is not quite complete."

--- "There is an issue with hyperparameter tuning in the experiments."


In particular, at least two reviewers mentioned the issues with different hyperparameters for different environments.

**Note From Pc:**

if the above contains the word "oral" or "spotlight" please see: "oral" presentation means -> notable-top-5% and "spotlight" means -> notable-top-25%. As stated in our emails, we are disassociating presentation type from AC recommendations